# Derivative-Free Guidance in Continuous and Discrete Diffusion Models with Soft Value-Based Decoding

## Abstract

Diffusion models excel at capturing the natural design spaces of images, molecules, and biological sequences of DNA, RNA, and proteins. However, for many applications from biological research to biotherapeutic discovery, rather than merely generating designs that are natural, we aim to optimize downstream reward functions while preserving the naturalness of these design spaces. Existing methods for achieving this goal often require "differentiable" proxy models (*e.g.*, classifier guidance) or computationally-expensive fine-tuning of diffusion models (*e.g.*, classifier-free guidance, RL-based fine-tuning). Here, we propose a new method, **S**oft Value-based **D**ecoding in **D**iffusion models (**SVDD**), to address these challenges. **SVDD** is an iterative sampling method that integrates soft value functions, which looks ahead to how intermediate noisy states lead to high rewards in the future, into the standard inference procedure of pre-trained diffusion models. Notably, **SVDD** avoids fine-tuning generative models and eliminates the need to construct differentiable models. This enables us to (1) directly utilize non-differentiable features/reward feedback, commonly used in many scientific domains, and (2) apply our method to recent discrete diffusion models in a principled way. Finally, we demonstrate the effectiveness of **SVDD** across several domains, including image generation, molecule generation (optimization of docking scores, QED, SA), and DNA/RNA generation (optimization of activity levels).

## 1 Introduction

Diffusion models have gained popularity as powerful generative models. Their applications extend beyond image generation to include natural language generation (Sahoo et al., 2024; Shi et al., 2024; Lou et al., 2023), molecule generation (Jo et al., 2022; Vignac et al., 2022), and biological (DNA, RNA, protein) sequence generation (Avdeyev et al., 2023; Stark et al., 2024). In each of these domains, diffusion models have been shown to be very effective at capturing complex natural distributions. However, in practice, we might not only want to generate *realistic* samples, but to produce samples that optimize specific downstream reward functions while preserving naturalness by leveraging pre-trained models. For example, in computer vision, we might aim to generate natural images with high aesthetic and alignment scores (Black et al., 2023; Fan et al., 2023). In drug discovery, we may seek to generate valid molecules with high QED/SA/docking scores (Lee et al., 2023; Jin et al., 2018) or RNAs (such as mRNA vaccines (Cheng et al., 2023)) with high translational efficiency and stability (Castillo-Hair and Seelig, 2021; Asrani et al., 2018), and regulatory DNAs that drives high cell-specificity of expression (Gosai et al., 2023; Taskiran et al., 2024; Lal et al., 2024).

The optimization of downstream reward functions using pre-trained diffusion models has been approached in various ways. In our work, we focus on non-fine-tuning-based methods because fine-tuning generative models (e.g., when using classifier-free guidance (Ho et al., 2020) or RL-based fine-tuning (Black et al., 2023; Fan et al., 2023; Uehara et al., 2024; Clark et al., 2023; Prabhudesai et al., 2023)) often becomes computationally intensive, especially as pre-trained generative models grow larger in the era of "foundation models". Although classifier guidance and its variants (*e.g.*, Dhariwal and Nichol (2021); Song et al. (2020); Chung et al. (2022); Bansal et al. (2023); Ho et al. (2022)) have shown some success as non-fine-tuning methods in these settings, they face significant

Table 1: A comparison of **SVDD** to prior approaches. "Non-differentiable" refers to the method's ability to operate without requiring differentiable proxy models. "No Training" means that no additional training of the diffusion model is required as long as we have access to the reward feedback. We compare to SMC-based methods in Section 6.

|  | No fine-tuning | Non-differentiable | No Training |
|---|:---:|:---:|:---:|
| Classifier guidance | ✓ | | |
| DPS (Chung et al., 2022) | ✓ | | ✓ |
| Classifier-free | | ✓ | |
| RL fine-tuning | | ✓ | |
| **SVDD**-MC (here) | ✓ | ✓ | |
| **SVDD**-PM (here) | ✓ | ✓ | ✓ |

challenges. First, as they would require constructing *differentiable* proxy models, they cannot directly incorporate useful *non-differentiable* features (*e.g.*, molecular/protein descriptors (van Westen et al., 2013; Ghiringhelli et al., 2015; Gainza et al., 2020)) or non-differentiable reward feedback (*e.g.*, physics-based simulations such as Vina and Rosetta (Trott and Olson, 2010; Alhossary et al., 2015; Alford et al., 2017)), which are particularly important in molecule design to optimize docking scores, stability, etc. This limitation also hinders the principled application of current classifier guidance methods to recently-developed discrete diffusion models (Austin et al., 2021; Campbell et al., 2022; Lou et al., 2023) (*i.e.*, without transforming the discrete space into the Euclidean space).

To tackle these challenges, we propose a novel method, **S**oft **V**alue-based **D**ecoding in **D**iffusion models (**SVDD**), for optimizing downstream reward functions in diffusion models (Figure 1). Inspired by recent literature on RL-based fine-tuning (Uehara et al., 2024), we first introduce soft value functions that serve as look-ahead functions, indicating how intermediate noisy samples lead to high rewards in the *future* of the diffusion denoising process. After learning (or approximating) these value functions, we present a new inference-time technique, **SVDD**, which obtains multiple noisy states from the policy (*i.e.*, denoising map) of pre-trained diffusion models and selects the sample with the highest value function at each time step. Specifically, we introduce two algorithms (**SVDD**-MC and **SVDD**-PM) depending on how we estimate value functions. Notably, the **SVDD**-PM approach does not require any additional learning as long as we have access to the

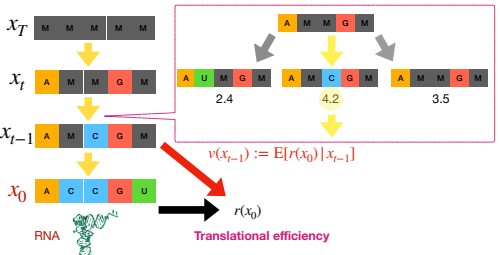

Figure 1: Summary of **SVDD**. $v$ denotes value functions that predict reward $r(x_0)$ (at time 0) from states at time $t-1$. **SVDD** involves two steps: (1) generating multiple noisy states from pre-trained models, and (2) selecting the state with the highest value according to the value function.

reward feedback by utilizing the characteristics of diffusion models (*i.e.,* the forward process in diffusion models to directly map $t$ to 0 in terms of expectation in Figure 1).

Our novel technique for optimizing downstream reward functions in pre-trained diffusion models makes two contributions (Table 1). First, it eliminates the need to construct differentiable proxy models. This allows for the use of non-differentiable reward feedback, which is common in many scientific fields, and makes our method applicable to recent discrete diffusion (Shi et al., 2024; Sahoo et al., 2024) models without any modification. Second, it avoids the need to fine-tune the generative model itself. This addresses the high computational cost associated with fine-tuning diffusion models. We demonstrate the effectiveness of our methods across diverse domains, including image generation, molecule generation (optimization of docking scores, QED, and SA), and DNA/RNA generation (optimization of activity levels).

## 2   RELATED WORKS

To summarize related work, we first outline methods relevant to our goal, categorizing them based on whether they involve fine-tuning. We then discuss related directions, such as discrete diffusion models, where our method excels. We defer other relevant works (e.g., decoding in LLMs) to Section A due to space constraints.

**Non-fine-tuning methods.** We discuss two main methods for optimizing rewards in diffusion models without fine-tuning. We further cover closely relevant methods based on sequential Monte Carlo (SMC) in Section 6 and Appendix B, after presenting our method.

- **Classifier guidance (Dhariwal and Nichol, 2021; Song et al., 2020):** It has been widely used to condition pre-trained diffusion models without fine-tuning. Although these methods do not originally focus on optimizing reward functions, they can be applied for this purpose (Uehara et al., 2024, Section 6.2). In this approach, an additional derivative of a certain value function is incorporated into the drift term (mean) of pre-trained diffusion models during inference. Subsequent variants (*e.g.*, Chung et al. (2022); Ho et al. (2022); Bansal et al. (2023); Guo et al. (2024); Wang et al. (2022); Yu et al. (2023)) have been proposed to simplify the learning of value functions. However, these methods require constructing *differentiable* models, which limits their applicability to non-differentiable features/reward feedbacks commonly encountered in scientific domains as mentioned in Section 1. Additionally, this approach cannot be directly extended to discrete diffusion models (e.g., (Lou et al., 2023; Shi et al., 2024; Sahoo et al., 2024)) in a principle way. Our approach aims to address these challenges.

  Note a notable exception of classifier guidance tailored to discrete diffusion models has been recently proposed by Nisonoff et al. (2024). However, **SVDD** can be applied to both continuous and discrete diffusion models in a unified manner. Furthermore, their practical method requires the differentiability of proxy models, unlike **SVDD**. We compare its performance with our method in Section 7. We provide further details in Section C.

- **Best-of-N:** The naive way is to generate multiple samples and select the top samples based on the reward functions, known as best-of-N in the literature on (autoregressive) LLMs (Stiennon et al., 2020; Nakano et al., 2021; Touvron et al., 2023; Beirami et al., 2024; Gao et al., 2023). This approach is significantly less efficient than ours, as **SVDD** uses soft-value functions that predict how intermediate noisy samples lead to high rewards in the future. We validate this experimentally in Section 7.

**Fine-tuning of diffusion models.** Several methods exist for fine-tuning generative models to optimize downstream reward functions, such as classifier-free guidance (Ho and Salimans, 2022) and RL-based fine-tuning (Fan et al., 2023; Black et al., 2023) or its variants (Dong et al., 2023; Wallace et al., 2024). However, these approaches often come with caveats, including high computational costs and the risk of easily forgetting pre-trained models. In our work, we propose an inference-time technique that eliminates the need for fine-tuning generative models.

**Discrete diffusion models.** Based on seminal works Austin et al. (2021); Campbell et al. (2022), recent work on masked diffusion models (Lou et al., 2023; Shi et al., 2024; Sahoo et al., 2024) has demonstrated their strong performance in natural language generation. Additionally, they have been applied to biological sequence generation (*e.g.*, DNA, protein sequences in Campbell et al. (2024); Sarkar et al. (2024)). In these cases, the use of diffusion models over autoregressive models is particularly apt, given that many biological sequences ultimately adopt complex three-dimensional structures. We also note that ESM3 (Hayes et al., 2024), a widely recognized foundation model in protein sequence generation, bears similarities to masked diffusion models. Despite its significance, it cannot be integrated with standard classifier guidance, because adding a continuous gradient to a discrete objective is not inherently valid. Unlike standard classifier guidance, **SVDD** can be seamlessly applied to discrete diffusion models.

## 3 PRELIMINARIES AND GOAL

We describe the standard method for training diffusion models and outline the objective of our work: optimizing downstream reward functions given pre-trained diffusion models.

### 3.1 DIFFUSION MODELS

In diffusion models (Ho et al., 2020; Song et al., 2020), our goal is to learn a sampler $p(x) \in \Delta(\mathcal{X})$ given data consisting of $x \in \mathcal{X}$. The training process for a standard diffusion model is summarized as follows. First, we introduce a (fixed) forward process $q_t : \mathcal{X} \to \Delta(\mathcal{X})$. Next, we aim to learn a backward process: $\{p_t\}$ where each $p_t$ is $\mathcal{X} \to \Delta(\mathcal{X})$ so that the distributions induced by the forward process and backward process match marginally. For this purpose, by parametrizing the backward

processes with $\theta \in \mathbb{R}^d$, we typically use the following loss function:

$$\mathbb{E}_{x_1,\cdots,x_T \sim q(\cdot|x_0)} \left[ -\log p_0(x_0|x_1) + \sum_{t=1}^{T} \mathrm{KL}(q_t(\cdot \mid x_{t-1})\|p_t(\cdot \mid x_{t+1};\theta)) + \mathrm{KL}(q_T(\cdot)\|p_T(\cdot)) \right],$$

which is derived from the variational lower bound of the negative log-likelihood (*i.e.*, ELBO).

Here are two examples of concrete parameterizations. Let $\alpha_t \in \mathbb{R}$ be a noise schedule.

**Example 1** (Continuous space)**.** *When $\mathcal{X}$ is Euclidean space, we typically use the Gaussian distribution $q_t(\cdot \mid x) = \mathcal{N}(\sqrt{\alpha_t}x, (1-\alpha_t))$. Then, the backward process is parameterized as*

$$\mathcal{N}\left( \frac{\sqrt{\alpha_t}(1-\bar{\alpha}_{t+1})x_t + \sqrt{\alpha_{t-1}}(1-\alpha_t)\hat{x}_0(x_t;\theta)}{1-\bar{\alpha}_t}, \frac{(1-\alpha_t)(1-\bar{\alpha}_{t-1})}{1-\bar{\alpha}_t} \right),$$

*where $\bar{\alpha}_t = \prod_{i=1}^{t} \alpha_i$. Here, $\hat{x}_0(x_t;\theta)$ is a neural network that predicts $x_0$ from $x_t$ ($\mathbb{E}_q[x_0|x_t]$).*

**Example 2** (Discrete space in Sahoo et al. (2024); Shi et al. (2024))**.** *Let $\mathcal{X}$ be a space of one-hot column vectors $\{x \in \{0,1\}^K : \sum_{i=1}^{K} x_i = 1\}$, and $\mathrm{Cat}(\pi)$ be the categorical distribution over $K$ classes with probabilities given by $\pi \in \Delta^K$ where $\Delta^K$ denotes the K-simplex. A typical choice is $q_t(\cdot \mid x) = \mathrm{Cat}(\alpha_t x + (1-\alpha_t)\mathbf{m})$ where $\mathbf{m} = [0,\cdots,0,\mathrm{Mask}]$. Then, the backward process is parameterized as*

$$\begin{cases} \mathrm{Cat}(x_t), & \text{if } x_t \neq \mathbf{m} \\ \mathrm{Cat}\left( \frac{(1-\alpha_{t-1})\mathbf{m}+(\alpha_{t-1}-\alpha_t)\hat{x}_0(x_t;\theta)}{1-\alpha_t} \right), & \text{if } x_t = \mathbf{m}. \end{cases}$$

*Here, $\hat{x}_0(x_t;\theta)$ is a neural network that predicts $x_0$ from $x_t$. Note that when considering a sequence of $L$ tokens ($x^{1:L}$), we use the direct product: $p_t(x_t^{1:L}|x_{t+1}^{1:L}) = \prod_{l=1}^{L} p_t(x_t^l|x_{t+1}^{1:L})$.*

After learning the backward process, we can sample from a distribution that emulates training data distribution (*i.e.*, $p(x)$) by sequentially sampling $\{p_t\}_{t=T}^{0}$ from $t=T$ to $t=0$.

**Notation.** The notation $\delta_a$ denotes a Dirac delta distribution centered at $a$. The notation $\propto$ indicates that the distribution is equal up to a normalizing constant. With slight abuse of notation, we often denote $p_T(\cdot|\cdot,\cdot)$ by $p_T(\cdot)$.

## 3.2 OBJECTIVE: GENERATING SAMPLES WITH HIGH REWARDS WHILE PRESERVING NATURALNESS

We consider a scenario where we have a pre-trained diffusion model, which is trained using the loss function explained in Section 3.1. These pre-trained models are typically designed to excel at characterizing the natural design space (*e.g.*, image space, biological sequence space, or chemical space) by emulating the extensive training dataset. Our work focuses on obtaining samples that also optimize downstream reward functions $r : \mathcal{X} \rightarrow \mathbb{R}$ (*e.g.*, Quantitative Estimate of Druglikeness (QED) and Synthetic Accessibility (SA) in molecule generation), while maintaining the naturalness by leveraging pre-trained diffusion models. We formalize this goal as follows.

Given a pre-trained model $\{p_t^{\mathrm{pre}}\}_{t=T}^{0}$, we denote the induced distribution by $p^{\mathrm{pre}} \in \Delta(\mathcal{X})$ (*i.e.*, $p^{\mathrm{pre}}(x_0) = \int\{\prod_{t=T+1}^{1} p_{t-1}^{\mathrm{pre}}(x_{t-1}|x_t)\}dx_{1:T}$). We aim to sample from the following distribution:

$$p^{(\alpha)}(x) := \underset{p \in [\Delta(\mathcal{X})]}{\mathrm{argmax}} \underbrace{\mathbb{E}_{x \sim p(\cdot)}[r(x)]}_{\textbf{term (a)}} - \alpha \underbrace{\mathrm{KL}(p(\cdot)\|p^{\mathrm{pre}}(\cdot))}_{\textbf{term(b)}} \propto \exp(r(x)/\alpha)p^{\mathrm{pre}}(x).$$

Here, term (a) is introduced to optimize the reward function, while term (b) is used to maintain the naturalness of the generated samples.

**Existing methods.** Several existing approaches target this goal (or its variant), as discussed in Section 2, including classifier guidance, fine-tuning (RL-based or classifier-free), and Best-of-N. In our work, we focus on non-fine-tuning-based methods; specifically, we aim to address the limitations of these methods: the requirement for differentiable proxy models in classifier guidance and the inefficiency of Best-of-N.

Finally, we note that all results discussed in this paper can be easily extended to cases where the pre-trained model is a conditional diffusion model. For example, in our image experiments (Section 7), the pre-trained model is a conditional diffusion model conditioned on text (e.g., Stable Diffusion).

## 4 SOFT VALUE-BASED DECODING IN DIFFUSION MODELS

First, we present the motivation behind developing our new algorithm. We then introduce **SVDD**, which satisfies our desired properties, *i.e.*, the lack of need for fine-tuning or constructing differentiable models.

### 4.1 KEY OBSERVATION

We introduce several key concepts. First, we define the *soft value function*:

$$t \in [T+1, \cdots, 1]; v_{t-1}(\cdot) := \alpha \log \mathbb{E}_{x_0 \sim p^{\mathrm{pre}}(x_0|x_{t-1})} \left[ \exp\left( \frac{r(x_0)}{\alpha} \right) | x_{t-1} = \cdot \right],$$

where $\mathbb{E}_{\{p^{\mathrm{pre}}\}}[\cdot]$ is induced by $\{p_t^{\mathrm{pre}}(\cdot|x_{t+1})\}_{t=T}^0$. This value function represents the expected future reward at $t = 0$ from the intermediate noisy state at $t - 1$.

Next, we define the following *soft optimal policy* (denoising process) $p_{t-1}^{\star,\alpha} : \mathcal{X} \to \Delta(\mathcal{X})$ weighted by value functions $v_{t-1} : \mathcal{X} \to \mathbb{R}$:

$$p_{t-1}^{\star,\alpha}(\cdot|x_t) = \frac{p_{t-1}^{\mathrm{pre}}(\cdot|x_t) \exp(v_{t-1}(\cdot)/\alpha)}{\int p_{t-1}^{\mathrm{pre}}(x|x_t) \exp(v_{t-1}(x)/\alpha) dx}.$$

Here, $v_t$ are soft value functions and $p_t^{\star,\alpha}$ are soft optimal policies, because they literally correspond, respectively, to soft value functions and soft optimal policies, where we embed diffusion models into entropy-regularized MDPs (Geist et al., 2019), as demonstrated in Uehara et al. (2024).

With this preparation in mind, we utilize the following key observation:

**Theorem 1** (From Theorem 1 in Uehara et al. (2024)). *The distribution induced by $\{p_t^{\star,\alpha}(\cdot|x_{t+1})\}_{t=T}^0$ is the target distribution $p^{(\alpha)}(x)$, i.e.,*

$$p^{(\alpha)}(x_0) = \int \left\{ \prod_{t=T+1}^1 p_{t-1}^{\star,\alpha}(x_{t-1}|x_t) \right\} dx_{1:T}.$$

While Uehara et al. (2024) presents this theorem, they use it primarily to interpret RL-based fine-tuning methods in Fan et al. (2023); Black et al. (2023). In contrast, our work explores how to convert this into a new fine-tuning-free optimization algorithm.

**Our motivation for a new algorithm.** Theorem 1 states that if we can hypothetically sample from $\{p_t^{\star,\alpha}\}_{t=T}^0$, we can sample from the target distribution $p^{(\alpha)}$. However, there are two challenges in sampling from each $p_t^{\star,\alpha}$: (1) the soft-value function $v_{t-1}$ in $p_{t-1}^{\star,\alpha}$ is unknown, and (2) it is unnormalized (*i.e.*, calculating the normalizing constant is hard).

We address the first challenge in Section 4.3. Assuming the first challenge is resolved, we consider how to tackle the second challenge. A natural approach is to use importance sampling (IS):

$$p_{t-1}^{\star,\alpha}(\cdot|x_t, c) \approx \sum_{m=1}^M \frac{w_{t-1}^{\langle m \rangle}}{\sum_{j=1}^M w_{t-1}^{\langle j \rangle}} \delta_{x_{t-1}^{\langle m \rangle}}, \quad \{x_{t-1}^{\langle m \rangle}\}_{m=1}^M \sim p_{t-1}^{\mathrm{pre}}(\cdot \mid x_t),$$

where $w_{t-1}^{\langle m \rangle} := \exp(v_t(x_{t-1}^{\langle m \rangle})/\alpha)$. Thus, we can approximately sample from $p_{t-1}^{\star,\alpha}(\cdot|x_t)$ by obtaining multiple ($M$) samples from pre-trained diffusion models and selecting the sample based on an index, which is determined by sampling from the categorical distribution with mean $\{w_{t-1}^{\langle m \rangle}/\sum_j w_{t-1}^{\langle j \rangle}\}_{m=1}^{\langle M \rangle}$.

Note that Best-of-N, which generates multiple samples and selects the highest reward sample, is technically considered IS, where the proposal distribution is the entire $p^{\mathrm{pre}}(x_0) = \int \prod_t \{p_t^{\mathrm{pre}}(x_{t-1} \mid x_t)\} dx_{1:T}$. However, the use of importance sampling in our algorithm differs significantly, as we apply it at each time step to approximate each soft-optimal policy.

### 4.2 INFERENCE-TIME ALGORITHM

Now, by leveraging the observation in Algorithm 1, we introduce our algorithm. Our algorithm is an iterative sampling method that integrates soft value functions into the standard inference procedure of pre-trained diffusion models. Each step is designed to approximately sample from a value-weighted policy $\{p_t^{\star,\alpha}\}_{t=T}^0$.

We note several key points.

---

**Algorithm 1 SVDD** (**S**oft **V**alue-Based **D**ecoding in **D**iffusion Models)

---

1: **Require**: Estimated soft value function $\{\hat{v}_t\}_{t=T}^{0}$ (refer to Algorithm 2 or Algorithm 3), pre-trained diffusion models $\{p_t^{\text{pre}}\}_{t=T}^{0}$, hyperparameter $\alpha \in \mathbb{R}$

2: **for** $t \in [T+1, \cdots, 1]$ **do**

3:     Get $M$ samples from pre-trained polices $\{x_{t-1}^{\langle m \rangle}\}_{m=1}^{M} \sim p_{t-1}^{\text{pre}}(\cdot|x_t)$, and for each $m$, calculate $w_{t-1}^{\langle m \rangle} := \exp(\hat{v}_{t-1}(x_{t-1}^{\langle m \rangle})/\alpha)$

4:     $x_{t-1} \leftarrow x_{t-1}^{\langle \zeta_{t-1} \rangle}$ after selecting an index: $\zeta_{t-1} \sim \text{Categorical}\left(\left\{\frac{w_{t-1}^{\langle m \rangle}}{\sum_{j=1}^{M} w_{t-1}^{\langle j \rangle}}\right\}_{m=1}^{M}\right),$

5: **end for**

6: **Output**: $x_0$

---

- When $\alpha = 0$, Line 4 corresponds to $\zeta_{t-1} = \text{argmax}_{m \in [1, \cdots, M]} \hat{v}_{t-1}(x_{t-1}^{\langle m \rangle})$. In practice, we often recommend this choice. This is the default choice in Section 7.

- A typical choice we recommend for $M$ is from 5 to 20. The performance with varying $M$ values will be discussed in Section 7.

- Line 3 can be computed in parallel at the cost of additional memory (scaled by $M$). If Line 3 is not computed in parallel, the computational time in SVDD would be approximately $M$ times that of the standard inference procedure. We will check it in Section 7.

- In special cases where the normalizing constant can be calculated relatively easily (*e.g.*, in discrete diffusion with small $K, L$), we can directly sample from $\{p_t^{\star, \alpha}\}_{t=T}^{0}$.

- A proposal distribution different from $p_{t-1}^{\text{pre}}$ in line 3 can be applied (see Section D). For instance, classifier guidance or its variants may be used to obtain better proposal distributions than those from the pure pre-trained model.

The remaining question is how to obtain the soft value function, which we address in the next section.

### 4.3 Learning Soft Value Functions

Next, we describe how to obtain soft value functions $v_t(x)$ in practice. We propose two main approaches: a Monte Carlo regression approach and a posterior mean approximation approach.

**Monte Carlo regression.** Here, we use the following approximation $v_t'$ as $v_t$ where

$$v_t'(\cdot) := \mathbb{E}_{x_0 \sim p^{\text{pre}}(x_0|x_t)}[r(x_0)|(x_t) = \cdot].$$

This is based on

$$v_t(x_t) = \alpha \log \mathbb{E}_{x_0 \sim p^{\text{pre}}(\cdot|x_t)}[\exp(r(x_0)/\alpha|x_t] \approx \log(\exp(\mathbb{E}_{x_0 \sim p^{\text{pre}}(x_t)}[r(x_0)|x_t])) = v_t'(x_t). \quad (1)$$

By regressing $r(x_0)$ onto $x_t$, we can learn $v_t'$ as in Algorithm 2. Combining this with Algorithm 1, we refer to the entire optimization approach as **SVDD**-MC.

---

**Algorithm 2** Value Function Estimation Using Monte Carlo Regression

---

1: **Require**: Pre-trained diffusion models, reward $r : \mathcal{X} \to \mathbb{R}$, function class $\Phi : \mathcal{X} \times [0, T] \to \mathbb{R}$.

2: Collect datasets $\{x_T^{(s)}, \cdots, x_0^{(s)}\}_{s=1}^{S}$ by rolling-out $\{p_t^{\text{pre}}\}_{t=T}^{0}$ from $t = T$ to $t = 0$.

3: $\hat{v}' = \text{argmin}_{f \in \Phi} \sum_{t=0}^{T} \sum_{s=1}^{S} \{r(x_0^{(s)}) - f(x_t^{(s)}, t)\}^2$.

4: **Output**: $\hat{v}'$

---

Note that technically, without the approximation introduced in (1), we can estimate $v_t$ by regressing $\exp(r(x_0)/\alpha)$ onto $x_t$ based on the original definition. This approach may work in many cases. However, when $\alpha$ is very small, the scaling of $\exp(r(\cdot)/\alpha)$ tends to be excessively large. Due to this concern, we generally recommend using Algorithm 2.

**Remark 1** (Another way of learning value functions). *Technically, another method for learning value functions is available such as soft-Q-learning (Section E), by leveraging soft-Bellman equations in diffusion models Uehara et al. (2024, Section 3) However, since we find Monte Carlo approaches to be more stable, we recommend them over soft-Q-learning.*

**Posterior mean approximation.** Here, recalling we use $\hat{x}_0(x_t)$ (approximation of $\mathbb{E}_{x_0 \sim p^{\mathrm{pre}}(x_t)}[x_0|x_t]$) when training pre-trained diffusion models in Section 3.1, we perform the following approximation:

$$v_t(x) := \alpha \log \mathbb{E}_{x_0 \sim p^{\mathrm{pre}}(x_0|x_t)}[\exp(r(x_0)/\alpha)|x_t] \approx \alpha \log(\exp(r(\hat{x}_0(x_t))/\alpha) = r(\hat{x}_0(x_t)).$$

Then, we can use $r(\hat{x}_0(x_t))$ as the estimated value function.

The advantage of this approach is that no additional training is required as long as we have $r$. When combined with Algorithm 1, we refer to the entire approach as **SVDD**-PM.

---

**Algorithm 3** Value Function Estimation using Posterior Mean Approximation

1: **Require**: Pre-trained diffusion models, reward $r : \mathcal{X} \to \mathbb{R}$
2: Set $\hat{v}^\diamond(\cdot, t) := r(\hat{x}_0(x_t = \cdot), t)$
3: **Output**: $\hat{v}^\diamond$

---

**Remark 2** (Relation with DPS). *In the context of classifier guidance, similar approximations have been employed (e.g., DPS in Chung et al. (2022)). However, the final inference-time algorithms differ significantly, as these methods compute gradients at the end.*

## 5 ADVANTAGES, EXTENSIONS, LIMITATIONS OF **SVDD**

We discuss the advantages, extensions, and limitations of **SVDD**.

### 5.1 ADVANTAGES

**No fine-tuning (or no training in SVDD-PM).** Unlike classifier-free guidance or RL-based fine-tuning, **SVDD** does not require any fine-tuning of the generative models. In particular, when using **SVDD**-PM, no additional training is needed as long as we have $r$.

**No need for constructing differentiable models.** Unlike classifier guidance, **SVDD** does not require differentiable proxy models, as there is no need for derivative computations. For example, if $r$ is non-differentiable feedback (*e.g.*, physically-based simulations for docking scores in molecule generation), our method **SVDD**-PM can directly utilize such feedback without constructing differentiable proxy models. In cases where non-differentiable computational feedback is costly to obtain, the usage of proxy reward models may still be preferred, but they do not need to be differentiable; thus, non-differentiable features or non-differentiable models based on scientific knowledge (*e.g.*, molecule fingerprints, GNNs) can be leveraged. Similarly, when using **SVDD**-MC, while a value function model is required, it does not need to be differentiable, unlike classifier guidance. Additionally, compared to approaches that involve derivatives (like classifier guidance or DPS), **SVDD** can be directly applied to discrete diffusion models mentioned in Example 2.

### 5.2 EXTENSIONS

**Using a likelihood/classifier as a reward.** While we primarily consider scenarios where reward models are regression models, by adopting a similar strategy to that in Zhao et al. (2024), they can be readily replaced with classifiers or likelihood functions in the context of solving inverse problems or conditioning (Chung et al., 2022; Bansal et al., 2023).

**Fine-tuning by distilling SVDD.** The inference-time cost may become slow as $M$ increases in **SVDD**. This issue can be mitigated by policy distillation, that is further fine-tuning diffusion models to align them closely with policies from **SVDD** (Salimans and Ho, 2022; Kim et al., 2023). We leave this aspect for future work.

### 5.3 POTENTIAL LIMITATIONS

**Memory and computational complexity in inference time.** Our approach requires more computational resources (if not parallelized) or memory (if parallelized), approximately $M$ times more than standard inference methods, as noted in Section 4.2. Taking this aspect into account, we compare **SVDD**, with baselines such as best-of-N in our experimental section (Section 7). For gradient-based approaches like classifier guidance and DPS, while a direct comparison with **SVDD** is challenging, it is important to note that these methods also incur additional computational and memory complexity due to the backward pass, which **SVDD** avoids. Lastly, it is important to note that this additional inference-time burden can be alleviated through distillation, as discussed in Section 5.2.

**Proximity to pre-trained models.** If significant changes to the pre-trained models are desired, we acknowledge that RL-based fine-tuning (Black et al., 2023; Fan et al., 2023) could be more effective than **SVDD** for this purpose in certain scenarios, such as image examples. However, this proximity to pre-trained models could also be advantageous in the sense that it is robust against reward optimization, which conventional fine-tuning methods often suffer from by exploiting these out-of-distribution regions (Uehara et al., 2024). Lastly, in cases where reward backpropagation (Prabhudesai et al., 2023; Clark et al., 2023; Uehara et al., 2024) is not applicable, particularly in scientific domains for RL-based fine-tuning, we may need to rely on PPO. However, PPO is often mode-seeking and unstable, highlighting the challenges of RL-based fine-tuning in certain scenarios.

## 6 COMPARISON BETWEEN SVDD AND SMC-BASED METHODS

SMC-based methods for diffusion models are closely related to **SVDD**. These approaches (Wu et al., 2024; Trippe et al., 2022; Dou and Song, 2024; Phillips et al., 2024; Cardoso et al., 2023) use SMC (Del Moral and Doucet, 2014) for sampling from diffusion models. While they are originally designed for conditioning (by setting rewards as classifiers), they can also be applied to reward maximization. Notably, similar to our work, these methods do not require differentiable models.

However, these SMC methods are not tailored to reward maximization. Most importantly, they involve resampling across the "entire" batch, which complicates parallelization. Additionally, when batch sizes are small, as is often the case with recent large diffusion models, performance may be suboptimal, since the SMC theoretical guarantees hold primarily with large batch sizes. Even with larger batch sizes, using SMC for reward maximization can result in a loss of diversity across the entire batch, since the effective sample size based on weights, which is a standard diversity measure in SMC, does not ensure "real" diversity in the generated samples. In contrast, our method is highly parallelizable, performs well even with small batch sizes (as low as 1), and maintains diversity with larger batch sizes, as sampling is conducted on a "per-sample basis" (Line 4). We empirically validate this in Section 7. We provide further details and experiments in Appendix B.

## 7 EXPERIMENTS

We conduct experiments to assess the performance of our algorithm relative to baselines and its sensitivity to various hyperparameters. We start by outlining the experimental setup, including baselines and models, and then present the results.

### 7.1 SETTINGS

**Methods to compare.** We compare **SVDD** to several representative methods capable of performing reward maximization during inference, discussed in Section 2.

- **Pre-trained models:** We generate samples using pre-trained models.

- **Best-of-N:** We generate samples from pre-trained models and select the top $1/N$ samples. This selection is made to ensure that the computational time during inference is approximately equivalent to that of **SVDD**.

- **DPS (Chung et al., 2022):** It is a widely used training-free version of classifier guidance. For discrete diffusion, we combine it with the state-of-the-art approach (Nisonoff et al., 2024).

- **SMC-Based Methods (SMC):** Methods discussed in Section 6 and Appendix B, which do not require differentiable models, like **SVDD**.

- **SVDD (Ours):** We implement **SVDD**-MC and **SVDD**-PM. We generally set $M = 20$ for images and $M = 10$ for other domains, and $\alpha = 0$. Recall $M$ is the duplication size in the IS part.

**Datasets and reward models.** We provide details on the pre-trained diffusion models and downstream reward functions used. For further information, refer to Section F.

- **Images**: We use Stable Diffusion v1.5 as the pre-trained diffusion model ($T = 50$). For downstream reward functions, we use compressibility and aesthetic scores (LAION Aesthetic Predictor V2 in Schuhmann (2022)), as employed by Black et al. (2023); Fan et al. (2023). Compressibility is a *non-differentiable reward feedback*.

- **Molecules**: We use GDSS (Jo et al., 2022), trained on ZINC-250k (Irwin and Shoichet, 2005), as the pre-trained diffusion model ($T = 1000$). For downstream reward functions, we use drug-likeness (QED) and synthetic accessibility (SA) calculated by RDKit, as well as docking score

(DS) calculated by QuickVina 2 (Alhossary et al., 2015), which are all *non-differentiable feedback*. Here, we renormalize SA to $(10 - \mathrm{SA})/9$ and docking score to $\max(-\mathrm{DS}, 0)$, so that a higher value indicates better performance. The docking scores measure binding affinity regarding four target proteins: Parp1, 5ht1b, Jak2, and Braf following Yang et al. (2021). These tasks are critical for drug discovery.

- **DNAs (enhancers) and RNAs (5'Untranslated regions (UTRs))**: We use the discrete diffusion model (Sahoo et al., 2024), trained on datasets from Gosai et al. (2023) for enhancers, and from Sample et al. (2019) for 5'UTRs, as our pre-trained diffusion model ($T = 128$). For the reward functions, we use an Enformer model (Avsec et al., 2021) to predict activity of enhancers in the HepG2 cell line, and a ConvGRU model that predicts the mean ribosomal load (MRL) of 5'UTRs measured by polysome profiling, respectively (Sample et al., 2019). These tasks are highly relevant for cell and RNA therapies, respectively (Taskiran et al., 2024; Castillo-Hair and Seelig, 2021).

## 7.2 RESULTS

Table 2: Top 10 and 50 quantiles of the generated samples for each algorithm (with 95% confidence intervals). Higher is better. **SVDD** consistently outperforms the baseline methods.

| Domain | Quantile | Pre-Train | Best-N | DPS | SMC | SVDD-MC | SVDD-PM |
|---|---|---|---|---|---|---|---|
| Image: Compress | 50% | -101.4 ± 0.22 | -71.2 ± 0.46 | -60.1 ± 0.44 | -59.7 ± 0.4 | -54.3 ± 0.33 | **-51.1** ± 0.38 |
| | 10% | -78.6 ± 0.13 | -57.3 ± 0.28 | -61.2 ± 0.28 | -49.9 ± 0.24 | -40.4 ± 0.2 | **-38.8** ± 0.23 |
| Image: Aesthetic | 50% | 5.62 ± 0.003 | 6.11 ± 0.007 | 5.61 ± 0.009 | 6.02 ± 0.004 | 5.70 ± 0.008 | **6.14** ± 0.007 |
| | 10% | 5.98 ± 0.002 | 6.34 ± 0.004 | 6.00 ± 0.005 | 6.28 ± 0.003 | 6.05 ± 0.005 | **6.47** ± 0.004 |
| Molecule: QED | 50% | 0.656 ± 0.008 | 0.835 ± 0.009 | 0.679 ± 0.024 | 0.667 ± 0.016 | **0.852** ± 0.011 | 0.848 ± 0.014 |
| | 10% | 0.812 ± 0.005 | 0.902 ± 0.006 | 0.842 ± 0.014 | 0.722 ± 0.009 | 0.925 ± 0.007 | **0.928** ± 0.008 |
| Molecule: SA | 50% | 0.652 ± 0.007 | 0.834 ± 0.014 | 0.693 ± 0.022 | 0.786 ± 0.004 | **0.935** ± 0.010 | 0.925 ± 0.016 |
| | 10% | 0.803 ± 0.004 | 0.941 ± 0.008 | 0.844 ± 0.013 | 0.796 ± 0.003 | **1.000** ± 0.006 | **1.000** ± 0.010 |
| Molecule: Docking parp1 | 50% | 7.15 ± 0.52 | 10.00 ± 0.17 | 7.35 ± 0.43 | 6.90 ± 0.60 | **12.00** ± 0.26 | 11.40 ± 0.22 |
| | 10% | 8.59 ± 0.31 | 10.67 ± 0.10 | 9.31 ± 0.26 | 9.37 ± 0.36 | **13.25** ± 0.16 | 12.41 ± 0.13 |
| Molecule: Docking 5ht1b | 50% | 7.20 ± 0.53 | 9.65 ± 0.17 | 7.30 ± 0.48 | 6.80 ± 0.35 | **10.50** ± 0.46 | **10.50** ± 0.53 |
| | 10% | 8.69 ± 0.32 | 10.28 ± 0.10 | 9.21 ± 0.29 | 9.00 ± 0.21 | **12.87** ± 0.28 | 12.30 ± 0.32 |
| Molecule: Docking jak2 | 50% | 7.05 ± 0.45 | 8.85 ± 0.17 | 7.30 ± 0.43 | 6.80 ± 0.46 | **10.65** ± 0.35 | 10.30 ± 0.45 |
| | 10% | 8.20 ± 0.27 | 9.59 ± 0.10 | 8.70 ± 0.26 | 10.00 ± 0.28 | 11.80 ± 0.21 | **11.91** ± 0.27 |
| Molecule: Docking braf | 50% | 7.20 ± 0.40 | 9.20 ± 0.11 | 7.50 ± 0.23 | 6.90 ± 0.46 | **10.00** ± 0.37 | 9.65 ± 0.33 |
| | 10% | 8.59 ± 0.24 | 10.29 ± 0.07 | 9.20 ± 0.14 | 8.74 ± 0.28 | 11.30 ± 0.22 | **11.40** ± 0.20 |
| Enhancers | 50% | 0.121 ± 0.033 | 1.807 ± 0.214 | 3.782 ± 0.299 | 4.28 ± 0.02 | 5.074 ± 0.096 | **5.353** ± 0.231 |
| | 10% | 1.396 ± 0.020 | 3.449 ± 0.128 | 4.879 ± 0.179 | 5.95 ± 0.01 | 5.639 ± 0.057 | **6.980** ± 0.138 |
| 5'UTR | 50% | 0.406 ± 0.028 | 0.912 ± 0.023 | 0.426 ± 0.073 | 0.76 ± 0.02 | 1.042 ± 0.008 | **1.214** ± 0.016 |
| | 10% | 0.869 ± 0.017 | 1.064 ± 0.014 | 0.981 ± 0.044 | 0.91 ± 0.01 | 1.117 ± 0.005 | **1.383** ± 0.010 |

We compare the baselines with our two proposed methods regarding rewards $r$ (Table 2). To demonstrate the generated samples' validity (i.e., naturalness), we present several examples in Figure 2. In Section F.3, we also include metrics for the validity of samples.

Overall, **SVDD** outperformed the baseline methods (**Best-of-N**, **SMC**, **DPS**, **SMC**), as evidenced by higher quantiles. Furthermore, for both molecules and images, the samples generated by **SVDD** were valid. This suggests that **SVDD** can generate high-reward valid samples that **Best-of-N**, **DPS**, and **SMC** often struggle to generate or, in some cases, nearly fail to do.

The relative performance of our **SVDD**-MC and **SVDD**-PM appears to be domain-dependent. Generally, **SVDD**-PM may be more robust since it does not require additional learning (*i.e.*, it directly utilizes reward feedback). The performance of **SVDD**-MC depends on the success of value function learning discussed in Section F.

**Ablation studies in terms of the duplication size $M$.** We assessed the performance of **SVDD**-PM (when calculating value functions in Line 3 in a non-parallel manner) along with the computational and memory complexity as $M$ varies. First, across all domains, the performance gradually plateaus as $M$ increases (Figure 3a and 3b). Second, computational complexity increases linearly with $M$, while memory complexity remains nearly constant (Figure 3c and 3d). This behavior is expected, as previously noted in Section 4.2. The comparison with **Best-of-N** in Table 2 is made with this consideration in mind.

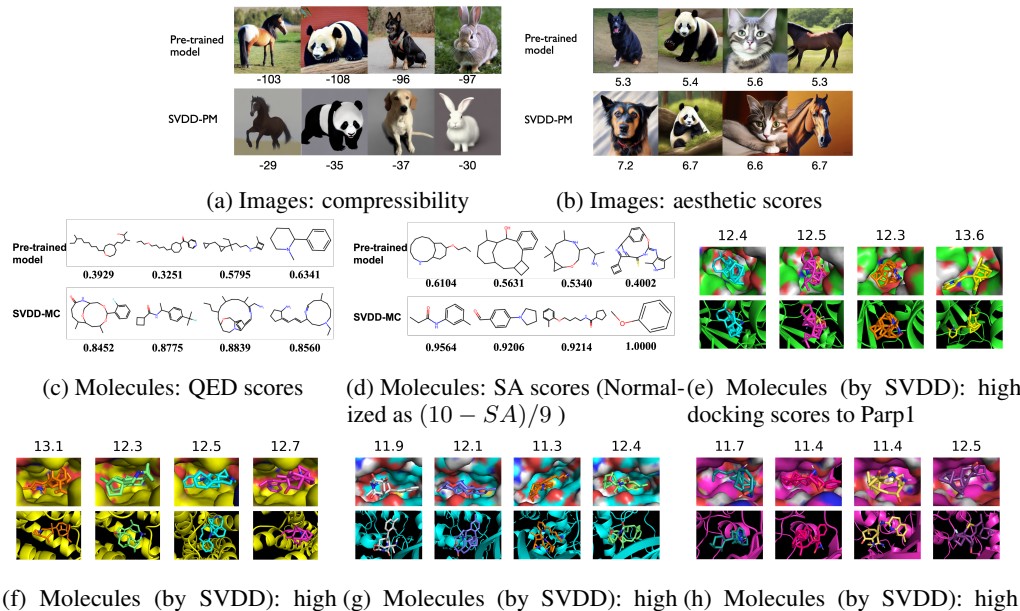

(a) Images: compressibility  (b) Images: aesthetic scores

(c) Molecules: QED scores  (d) Molecules: SA scores (Normal-  (e) Molecules (by SVDD): high ized as $(10 - SA)/9$ )  docking scores to Parp1

(f) Molecules (by SVDD): high docking scores to 5ht1b  (g) Molecules (by SVDD): high docking scores to Jak2  (h) Molecules (by SVDD): high docking scores to Braf

Figure 2: Generated samples from **SVDD**. For more samples, please refer to Section F.3. Note that the surfaces and ribbons in (e)-(h) (such as the green objects in (e)) are representations of the target proteins, while the generated small molecules are displayed in the center.

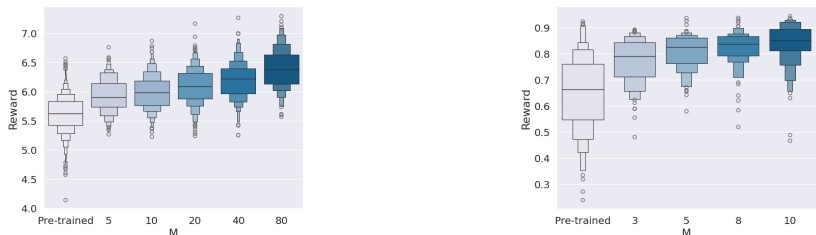

(a) Performance of **SVDD** as $M$ varies for image generation while optimizing aesthetic score.  (b) Performance of **SVDD** as $M$ varies for molecule generation while optimizing the QED score.

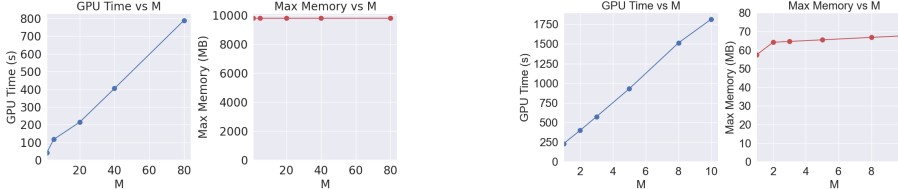

(c) GPU time and max memory of **SVDD** as $M$ varies for image generation (aesthetic scores).  (d) GPU time and max memory of **SVDD** as $M$ varies for molecule generation (QED).

Figure 3: Ablation studies with respect to $M$ for **SVDD**. Note 3c and 3d indicate that the computational time does not scale linearly with $M$, whereas memory usage scales linearly.

## 8 CONCLUSION

We propose a novel inference-time algorithm, **SVDD**, for optimizing downstream reward functions in pre-trained diffusion models that eliminate the need to construct differentiable proxy models. Future works include applications in other domains, such as protein sequence optimization (Gruver et al., 2023; Alamdari et al., 2023; Watson et al., 2023) and 3D molecule generation (Xu et al., 2023).

## REPRODUCIBILITY STATEMENT

To ensure the reproducibility of this work, we provide full details for the experiments including all the training setup, architecture, hyper-parameter searching spaces and metrics in Appendix F. We also provide an anonymous code link containing the implementation of our method and baselines: `https://anonymous.4open.science/r/SVDD/`.

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

## A  FURTHER RELATED WORKS

**Decoding in autoregressive models with rewards.**   The decoding strategy, which dictates how sentences are generated from the model, is a critical component of text generation in autoregressive language models (Wu et al., 2016; Chorowski and Jaitly, 2016; Leblond et al., 2021). Recent studies have explored inference-time techniques for optimizing downstream reward functions Dathathri et al. (2019); Yang and Klein (2021); Qin et al. (2022); Mudgal et al. (2023); Zhao et al. (2024); Han et al. (2024). While there are similarities between these works and ours, to the best of our knowledge, no prior work has extended such methodologies to diffusion models. Furthermore, our approach leverages characteristics unique to diffusion models that are not present in autoregressive models such as **SVDD**-PM.

## B  DIFFERENCE BETWEEN **SVDD** AND "STANDARD" SMC-METHODS

In this section, we compare our algorithm with the SMC-based methods (Wu et al., 2024; Trippe et al., 2022; Dou and Song, 2024; Phillips et al., 2024; Cardoso et al., 2023) for guidance. While they originally aim to solve a conditioning problem, which is different from reward maximization, they algorithms can be converted to reward maximization. First, we will explain this converted algorithm. We then elaborate on the differences between **SVDD**, and them in the context of reward maximization. Notably, our SVDD is an instantiation of nested-IS SMC (Naesseth et al., 2019, Algorithm 5) in the literature on computational statistics, whereas these SMC-based Methods rely on standard sequential Monte Carlo.

### B.1  SMC-BASED METHODS

---

**Algorithm 4** Guidance with "Standard" SMC (for reward maximization)

---

1: **Require**: Estimated value functions $\{\hat{v}_t(x)\}_{t=T}^0$, pre-trained diffusion models $\{p_t^{\mathrm{pre}}\}_{t=T}^0$, hyper-parameter $\alpha \in \mathbb{R}$, Batch size $N$

2: **for** $t \in [T+1, \cdots, 0]$ **do**

3:   **IS step:**

4:

$$i \in [1, \cdots, N]; x_{t-1}^{[i]} \sim p_{t-1}^{\mathrm{pre}}(\cdot | x_t^{[i]}), w_{t-1}^{[i]} := \frac{\exp(\hat{v}_{t-1}(x_{t-1}^{[i]})/\alpha)}{\exp(\hat{v}_t(x_t^{[i]})/\alpha)}$$

5:   **Selection step:** select new indices with replacement

6:   $\{x_{t-1}^{[i]}\}_{i=1}^N \leftarrow \{x_{t-1}^{\zeta_{t-1}^{[i]}}\}_{i=1}^N, \quad \{\zeta_{t-1}^{[i]}\}_{i=1}^N \sim \mathrm{Cat}\left(\left\{\frac{w_{t-1}^{[i]}}{\sum_{j=1}^N w_{t-1}^{[j]}}\right\}_{i=1}^N\right)$

7: **end for**

8: **Output**: $x_0$

---

The complete algorithm of TDS in our setting is summarized in Algorithm 4. Since our notation and their notations are slightly different, we first provide a brief overview. It consists of two steps. Since our algorithm is iterative, at time point $t$, consider we have $N$ samples (particles) $\{x_t^{[i]}\}_{i=1}^N$.

**IS step (line 3).**   We generate a set of samples $\{x_{t-1}^{[i]}\}_{i=1}^N$ following a policy from a pre-trained model $p_{t-1}^{\mathrm{pre}}(\cdot|\cdot)$. In other words,

$$\forall i \in [1, \cdots, N]; x_{t-1}^{[i]} \sim p_{t-1}^{\mathrm{pre}}(\cdot | x_t^{[i]}).$$

Now, we denote the importance weight for the next particle $x_{t-1}$ given the current particle $x_t$ as $w(x_{t-1}, x_t)$, expressed as

$$w(x_{t-1}, x_t) := \frac{\exp(v_{t-1}(x_{t-1})/\alpha)}{\int \exp(v_{t-1}(x_{t-1})/\alpha) p_{t-1}^{\mathrm{pre}}(x_{t-1}|x_t) dx_{t-1}} = \frac{\exp(v_{t-1}(x_{t-1})/\alpha)}{\exp(v_t(x_t)/\alpha)},$$

and define
$$\forall i \in [1, \cdots, N]; \quad w_{t-1}^{[i]} := w(x_{t-1}^{[i]}, x_t^{[i]}).$$
Note here we have used the soft Bellman equation:

$$\exp(v_t(x_t)/\alpha) = \int \exp(v_{t-1}(x_{t-1})/\alpha) p_{t-1}^{\text{pre}}(x_{t-1}|x_t) dx_{t-1}.$$

Hence, by denoting the target marginal distribution at $t-1$, we have the following approximation:

$$p_{t-1}^{\text{tar}} \underset{\text{IS}}{\approx} \sum_{i=1}^{N} \frac{w_{t-1}^{[i]}}{\sum_{j=1}^{N} w_{t-1}^{[j]}} \delta_{x_{t-1}^{[i]}}.$$

**Selection step (line 5).** Finally, we consider a resampling step. The resampling indices are determined by the following:

$$\{\zeta_{t-1}^{[i]}\}_{i=1}^{N} \sim \text{Cat}\left( \left\{ \frac{w_{t-1}^{[i]}}{\sum_{j=1}^{N} w_{t-1}^{[j]}} \right\}_{i=1}^{N} \right).$$

To summarize, we conduct

$$p_{t-1}^{\text{tar}} \underset{\text{IS}}{\approx} \sum_{i=1}^{N} \frac{w_{t-1}^{[i]}}{\sum_{j=1}^{N} w_{t-1}^{[j]}} \delta_{x_{t-1}^{[i]}} \underset{\text{Resampling}}{\approx} \frac{1}{N} \sum_{i=1}^{N} \delta_{x_{t-1}^{\zeta_{t-1}^{[i]}}}.$$

Finally, we give several important remarks.

- In SMC, resampling is performed across the *entire* batch. However, in the algorithm, sampling is done within a single batch. Therefore, the algorithms differ significantly. We will discuss the implications in the next section.
- All of existing works Wu et al. (2024); Cardoso et al. (2023); Phillips et al. (2024); Dou and Song (2024) actually consider a scenario where the reward $r$ is a classifier. In Algorithm 4, we tailor the algorithm for reward maximization. Vice verisa, as we mentioned in Section 5.2, our **SVDD** can also operate effectively when $r$ is a classifier.
- In Wu et al. (2024); Cardoso et al. (2023); Phillips et al. (2024), the proposal distribution is not limited to the pre-trained model. Likewise, in our **SVDD**, we can select an arbitrary proposal distribution, as discussed in Section D.
- In the context of autoregressive (language) models, Zhao et al. (2024); Lew et al. (2023) proposed a similar algorithm.

## B.2 COMPARISON OF SVDD WITH "STANDARD" SMC-BASED METHODS (SSM) FOR REWARD MAXIMIZATION

We now compare our **SVDD** with "standard" SMC-Based Methods (SSM). Here, we write a batch size of SVDD in $G$. Importantly, we note that our implementation is analogous to nested-IS SMC in the literature in computational statistics; hence, many differences between nested-IS SMC (Naesseth et al., 2019; 2015) and pure SMC in computational statistics are translated here.

We first reconsider the fundamental assumptions of each algorithm. SVDD's performance, in terms of rewards, depends on the size of $M$ but is independent of the batch size $G$. In contrast, the performance of SSM depends on the batch size $N$. With this in mind, we compare the advantages of SVDD over SSM from various perspectives.

**Tailored to optimization in SVDD.** **SVDD** is considered more suitable for optimization than SMC. This is because, when using SMC for reward maximization, we must set $\alpha$ very low, leading to a lack of diversity. This is expected, as when $\alpha$ approaches 0, the effective sample size reduces to 1. This effect is also evident in our image experiments, as shown in Figure 4. Although SMC performs well in terms of reward functions, there is a significant loss of diversity. Some readers might think this could be mitigated by calculating the effective sample size based on weights (i.e., value functions) and resampling when the effective size decreases; however, this is not the case, as the effective sample size does not directly translate into greater diversity in the generated samples. In contrast, **SVDD**, maintains much higher diversity.

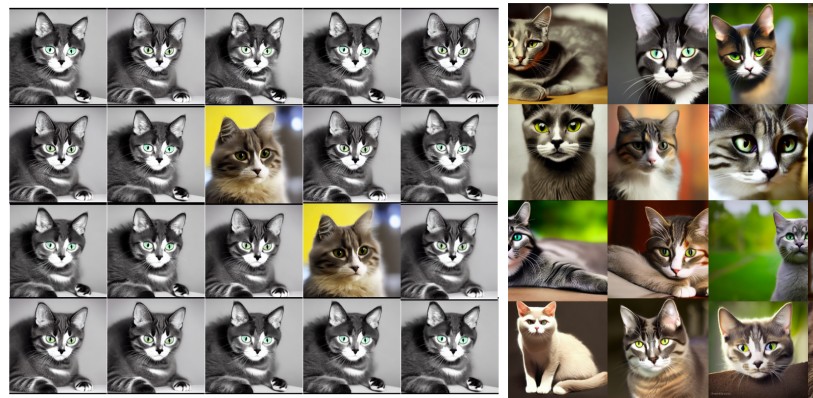

(a) Samples from SMC        (b) Samples from **SVDD**-PM

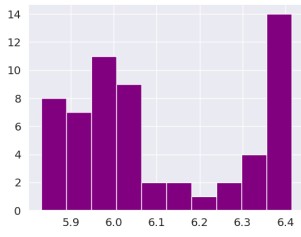
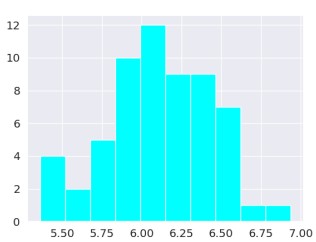

(c) Aesthetic scores from SMC        (d) Aesthetic scores from **SVDD**-PM

Figure 4: Examples of generated samples from SMC (left) and **SVDD** (right), when the prompt is "cat." The histogram of generated samples in terms of the reward function is shown below. Here, the pre-trained models are based on stable diffusion, and we optimize for aesthetic scores. In SMC, we set the batch size $N = 60$, and in **SVDD**, we set the duplication size $M = 20$. It is observed that most samples generated by SMC are similar, although diversity in terms of reward functions is roughly maintained. This suggests that the effective sample size in terms of value functions (i.e., weights) does not directly translate to real diversity in the generated samples. On the other hand, in SVDD, the generated samples are much more diverse while still achieving high reward functions.

**Ease of parallelization in SVDD.** SVDD is significantly easier to parallelize across multiple nodes. In contrast, SSM requires interaction between nodes for parallelization. This advantage is also well-documented in the context of nested-IS SMC versus standard SMC (Naesseth et al., 2019).

**High performance of SVDD under memory constraints.** Now, consider a scenario where the batch size is small, which often occurs due to memory constraints in large pre-trained diffusion models. In this case, while SSM may exhibit suboptimal performance, SVDD-PM can still achieve high performance by choosing a sufficiently large $M$.

**In SSM, the "ratio" is approximated.** In SVDD, we approximate each $\exp(v_{t-1}(x_{t-1})/\alpha)$ as a weight. However, in standard SMC, the ratio is approximated as a weight:

$$\frac{\exp(v_{t-1}(x_{t-1})/\alpha)}{\exp(v_t(x_t)/\alpha)}.$$

The key difference is that in SSM, both the numerator and the denominator are approximated, which could lead to greater error propagation.

## C    COMPARISON AGAINST DG (NISONOFF ET AL., 2024) AND DiGRESS (VIGNAC ET AL., 2022) IN DISCRETE DIFFUSION MODELS

Our method is closely related to guidance methods used in DG (Nisonoff et al., 2024) and DiGress (Vignac et al., 2022). However, we emphasize that our approach is more general, as it operates on any

domain, including continuous spaces including Riemannian spaces and discrete spaces, in a unified manner. In this sense, a strict comparison is not feasible. With this in mind, we provide a comparison focusing on cases where all domains are discrete.

**Comparison with Nisonoff et al. (2024).** In this continuous framework, following the notation in (Lou et al., 2023), they propose the use of the following rate matrix:

$$Q_{x,y}^{\star}(t) = Q_{x,y}^{\mathrm{pre}}(t)\frac{\exp(v_t(y)/\alpha)}{\exp(v_t(x)/\alpha)}.$$

where $Q_{x,y}^{\mathrm{pre}}(t)$ is a rate matrix in the pre-trained model. This suggests that, with standard discretization, the optimal policy at each time step is:

$$p(x_{t+\delta t} = y | x_t = x) = \mathrm{I}(x \neq y) + Q_{x,y}^{\star}(t)(\delta t) \tag{2}$$

where $\delta t$ is step size. Asymptotically, this is equivalent to sampling from the optimal policy in Theorem 1, as we will show in Remark 3. However, as Nisonoff et al. (2024) note, sampling from the optimal policy requires $O(KL)$ computation, where $K$ is the vocabulary size and $L$ is the sequence length, which is computationally expensive for large $K$ and $L$. To address this issue, they propose using a Taylor approximation by computing the gradient once. However, this is a heuristic in the sense that there is no theoretical guarantee for this approximation. In contrast, we avoid this computational overhead in a different manner, i.e., through importance sampling and resampling. Our algorithm has an asymptotic guarantee as $M$ goes to infinity. Empirically, we have compared two methods in Section 7.

**Remark 3** (Asymptotic equivalence between formula in Nisonoff et al. (2024) and Theorem 1 )**.** *The informal reasoning is as follows. Recall that the pre-trained policy can be written as*

$$p(x_{t+\delta t} = y | x_t = x) = \mathrm{I}(x \neq y) + Q_{x,y}^{\mathrm{pre}}(t)(\delta t).$$

*Then, Theorem 1 states that the optimal policy is*

$$\frac{\{\mathrm{I}(x \neq y) + Q_{x,y}^{\mathrm{pre}}(t)(\delta t)\}\exp(v_t(y)/\alpha)}{\sum_z \{\mathrm{I}(x \neq z) + Q_{x,z}^{\mathrm{pre}}(t)(\delta t)\}\exp(v_t(z)/\alpha)}.$$

*Now, we have*

$$\frac{\{\mathrm{I}(x \neq y) + Q_{x,y}^{\mathrm{pre}}(t)(\delta t)\}\exp(v_t(y)/\alpha)}{\sum_z \{I(x \neq z) + Q_{x,z}^{\mathrm{pre}}(t)(\delta t)\}\exp(v_t(z)/\alpha)}$$

$$= \frac{\mathrm{I}(x \neq y) + Q_{x,y}^{\mathrm{pre}}(t)(\delta t)\}\exp(v_t(y)/\alpha)}{\exp(v_t(x)/\alpha)} \times \{1 + O(\delta t)\}$$

$$\approx \frac{\{\mathrm{I}(x \neq y) + Q_{x,y}^{\mathrm{pre}}(t)(\delta t)\}\exp(v_t(y)/\alpha)}{\exp(v_t(x)/\alpha)} = \mathrm{I}(x \neq y) + \frac{Q_{x,y}^{\mathrm{pre}}(t)(\delta t)\exp(v_t(y)/\alpha)}{\exp(v_t(x)/\alpha)}.$$

*Thus, this recovers the formula* (2).

**Comparison with DiGress in Vignac et al. (2022).** Vignac et al. (2022) proposed a diffusion model for graphs where each sampling and denoising step operates directly on the discrete structure, avoiding continuous relaxation. They discuss how to implement guidance by treating rewards as a classifier. To bypass the exponential computational cost of sampling from the optimal policy ($p^{(\alpha)}$ in Theorem 1), they employ a Taylor expansion. While it requires the calculation of gradients for value functions, then it mitigates the exponential blow-up in computational time. In contrast, we avoid this computational blow-up through importance sampling (IS) and resampling. A detailed empirical comparison between our method and theirs is left for future work.

**Remark 4.** *Note that in our molecule generation experiment in Section 7, we use GDSS (Jo et al., 2022), which operates in continuous space and differs from DiGress.*

## D  EXTENSION WITH ARBITRARY PROPOSAL DISTRIBUTION

Here, we describe the algorithm where the proposal distribution is not necessarily derived from the policy of the pre-trained model, as summarized in Algorithm 5. Essentially, we only adjust the importance weight. In practice, we can use the gradient of a differentiable proxy model, such as DPS, as the proposal distribution $q_{t-1}$. Even if the differentiable proxy (value function) models are not highly accurate, our method will still perform effectively since other value function models $\hat{v}_{t-1}$ can be non-differentiable.

---

**Algorithm 5 SVDD** (**S**oft **V**alue-Based **D**ecoding in **D**iffusion Models)

---

1: **Require**: Estimated soft value function $\{\hat{v}_t\}_{t=T}^0$ (refer to Algorithm 2 or Algorithm 3), pre-trained diffusion models $\{p_t^{\text{pre}}\}_{t=T}^0$, hyperparameter $\alpha \in \mathbb{R}$, proposal distribution $\{q_t\}_{t=T}^0$

2: **for** $t \in [T+1, \cdots, 1]$ **do**

3:      Get $M$ samples from pre-trained polices $\{x_{t-1}^{\langle m \rangle}\}_{m=1}^M \sim q_{t-1}(\cdot|x_t)$, and for each $m$, and calculate

$$w_{t-1}^{\langle m \rangle} := \exp(\hat{v}_{t-1}(x_{t-1}^{\langle m \rangle})/\alpha) \times \frac{p_{t-1}^{\text{pre}}(x_{t-1}^{\langle m \rangle}|x_t)}{q_{t-1}(x_{t-1}^{\langle m \rangle}|x_t)}.$$

4:      $x_{t-1} \leftarrow x_{t-1}^{\langle \zeta_{t-1} \rangle}$ after selecting an index: $\zeta_{t-1} \sim \text{Cat}\left(\left\{\frac{w_{t-1}^{\langle m \rangle}}{\sum_{j=1}^M w_{t-1}^{\langle j \rangle}}\right\}_{m=1}^M\right),$

5: **end for**

6: **Output**: $x_0$

---

# E    SOFT Q-LEARNING

In this section, we explain soft value iteration to estimate soft value functions, which serves as an alternative to Monte Carlo regression.

**Soft Bellman equation.**    Here, we use the soft Bellman equation:

$$\exp(v_t(x_t)/\alpha) = \int \exp(v_{t-1}(x_{t-1})/\alpha)p_{t-1}^{\text{pre}}(x_{t-1}|x_t)dx_{t-1},$$

as proved in Section 4.1 in (Uehara et al., 2024). In other words,

$$v_t(x_t) = \alpha \log\{\mathbb{E}_{x_{t-1} \sim p^{\text{pre}}(\cdot|x_t)}[\exp(v_{t-1}(x_{t-1})/\alpha)|x_t]\}.$$

**Algorithm.**    Based on the above, we can estimate soft value functions recursively by regressing $v_{t-1}(x_{t-1})$ onto $x_t$. This approach is often referred to as soft Q-learning in the reinforcement learning literature (Haarnoja et al., 2017; Levine, 2018).

---

**Algorithm 6** Value Function Estimation Using Soft Q-learning

---

1: **Require**: Pre-trained diffusion models $\{p_t^{\text{pre}}\}_{t=T}^0$, value function model $v(x; \theta)$

2: Collect datasets $\{x_T^{(s)}, \cdots, x_0^{(s)}\}_{s=1}^S$ by rolling-out $\{p_t^{\text{pre}}\}_{t=T}^0$ from $t = T$ to $t = 0$.

3: **for** $j \in [0, \cdots, J]$ **do**

4:      Update $\theta$ by running regression:

$$\theta_j' \leftarrow \underset{\theta}{\text{argmin}} \sum_{t=0}^T \sum_{s=1}^S \left\{v(x_t^{(s)}; \theta) - v(x_{t-1}^{(s)}; \theta_{j-1}')\right\}^2.$$

5: **end for**

6: **Output**: $v(x; \theta_J')$

---

In our context, due to the concern of scaling of $\alpha$, as we have done in Algorithm 2, we had better use

$$v_t(x_t) = \mathbb{E}_{x_{t-1} \sim p^{\text{pre}}(\cdot|x_t)}[v_{t-1}(x_{t-1})|x_t].$$

With the above recursive equation, we can estimate soft value functions as in Algorithm 6.

# F    ADDITIONAL EXPERIMENTAL DETAILS

We further add additional experimental details.

### F.1 ADDITIONAL SETUPS FOR EXPERIMENTS

#### F.1.1 SETTINGS

**Images.** We define compressibility score as the negative file size in kilobytes (kb) of the image after JPEG compression following (Black et al., 2023). We define aesthetic scorer implemented as a linear MLP on top of the CLIP embeddings, which is trained on more than 400k human evaluations. As pre-trained models, we use Stable Diffusion, which is a common text-to-image diffusion model. As prompts to condition, we use animal prompts following (Black et al., 2023) such as [Dog, Cat, Panda, Rabbit, Horse,...].

**Molecules.** We calculate QED and SA scores using the RDKit (Landrum et al., 2016) library. We use the docking program QuickVina 2 (Alhossary et al., 2015) to compute the docking scores following Yang et al. (2021), with exhaustiveness as 1. Note that the docking scores are initially negative values, while we reverse it to be positive and then clip the values to be above 0, *i.e.*. We compute DS regarding four proteins, parp1 (Poly [ADP-ribose] polymerase-1), 5ht1b (5-hydroxytryptamine receptor 1B), braf (Serine/threonine-protein kinase B-raf), and jak2 (Tyrosine-protein kinase JAK2), which are target proteins that have the highest AUROC scores of protein-ligand binding affinities for DUD-E ligands approximated with AutoDock Vina.

**DNA, RNA sequences.** We examine two publicly available large datasets: enhancers ($n \approx 700k$) (Gosai et al., 2023) and UTRs ($n \approx 300k$) (Sample et al., 2019), with activity levels measured by massively parallel reporter assays (MPRA) (Inoue et al., 2019). These datasets have been widely used for sequence optimization in DNA and RNA engineering, particularly in advancing cell and RNA therapies (Castillo-Hair and Seelig, 2021; Lal et al., 2024; Ferreira DaSilva et al., 2024; Uehara et al., 2024). In the Enhancers dataset, each $x$ is a DNA sequence of length 200, while $y \in \mathbb{R}$ is the measured activity in the Hep cell line. In the 5'UTRs dataset, $x$ is a 5'UTR RNA sequence of length 50, and $y \in \mathbb{R}$ is the mean ribosomal load (MRL) measured by polysome profiling.

#### F.1.2 BASELINES AND PROPOSALS

We will explain in more detail how to implement baselines and our proposal. We use A100 GPUs for all the tasks.

**SVDD-MC.** In SVDD-MC, we require value function models. For images, we use standard CNNs for this purpose, with the same architecture as the reward model. For molecular tasks, we use a Graph Isomorphism Network (GIN) model (Xu et al., 2018) as the value function model. Notably, this model is not differentiable w.r.t. inputs. For GIN, we use mean global pooling and the RELU activation function, and the dimension of the hidden layer is 300. The number of convolutional layers in the GIN model is selected from the set $\{3, 5\}$; and we select the maximum number of iterations from $\{300, 500, 1000\}$, the initial learning rate from $\{1e-3, 3e-3, 5e-3, 1e-4\}$, and the batch size from $\{32, 64, 128\}$. For the Enhancer task, we use the Enformer model (Avsec et al., 2021) as the value function model. The Enformer trunk has 7 convolutional layers, each having 1536 channels. as well as 11 transformer layers, with 8 attention heads and a key length of 64. Dropout regularization is applied across the attention mechanism, with an attention dropout rate of 0.05, positional dropout of 0.01, and feedforward dropout of 0.4. The convolutional head for final prediction has 2*1536 input channels and uses average pooling, without an activation function. The model is trained using a batch size selected from $\{32, 64, 128, 256\}$, the learning rate from $\{1e-4, 5e-4, 1e-3\}$, and the maximum number of iterations from $\{5k, 10k, 20k\}$. For the 5'UTR task, we adopt the ConvGRU model (Dey and Salem, 2017). The ConvGRU trunk has a stem input with 4 channels and a convolutional stem that outputs 64 channels using a kernel size of 15. The model contains 6 convolutional layers, each initialized with 64 channels and a kernel size of 5. The convolutional layers use ReLU as the activation function, and a residual connection is applied across layers. Batch normalization is applied to both the convolutional and GRU layers. A single GRU layer with dropout of 0.1 is added after the convolutional layers. The convolutional head for final prediction uses 64 input channels and average pooling, without batch normalization. For training, the batch size is selected from $\{16, 32, 64, 128\}$, the learning rate from $\{1e-4, 2e-4, 5e-4\}$, and the maximum number of iterations from $\{2k, 5k, 10k\}$. All function models are trained to converge in the learning process using MSE loss.

**SVDD-PM.** For this proposal, we directly use the reward feedback to evaluate. We remark when the reward feedback is also learned from offline data, technically, it would be better to use techniques mitigating over-optimization as discussed in Uehara et al. (2024). However, since this point is tangential in our work, we don't do it.

**DPS.** We require differentiable models. For this task, for images, enhancers, and 5'UTRs, we use the same method as SVDD-MC. For molecules, we follow the implementation in Lee et al. (2023), and we use the same GNN model as the reward model. Note that we cannot compute derivatives with respect to adjacency matrices when using the GNN model. Regarding $\alpha$, we choose several candidates and report the best one. For image tasks we select from [5.0, 10.0] and for bio-sequence tasks we select from [1.0, 2.0]. For molecule QED task we select from {0.2, 0.3, 0.4, 0.5}, for molecule SA task {0.1, 0.2, 0.3}, and for molecule docking tasks we select from {0.4, 0.5, 0.6}.

**SMC.** For value function models, we use the same method as **SVDD**-PM. Regarding $\alpha$, we choose several candidates and report the best one. For image tasks we select from [10.0, 40.0]. For Enhancer and 5'UTR tasks as well as molecule QED and SA tasks we select from {0.1, 0.2, 0.3, 0.4}, while for molecule docking tasks we select from {1.5, 2.0, 2.5}.

### F.2 SOFTWARE AND HARDWARE

Our implementation is under the architecture of PyTorch (Paszke et al., 2019). The deployment environments are Ubuntu 20.04 with 48 Intel(R) Xeon(R) Silver, 4214R CPU @ 2.40GHz, 755GB RAM, and graphics cards NVIDIA RTX 2080Ti. Each of our experiments is conducted on a single NVIDIA RTX 2080Ti or RTX A6000 GPU.

### F.3 ADDITIONAL RESULTS

**Histograms.** In the main text, we present several quantiles. Here, we plot the reward score distributions of generated samples as histograms in Figure 5.

**Performance of value function training.** We report the performance of value function learning using Monte Carlo regression as follows in **SVDD**-MC. In Figure 6, we plot the Pearson correlation on the test dataset for the Enhancer and 5'UTR tasks, as well as the test MSE for the molecular task of parp1 docking score.

**Validity metrics for molecule generation.** To evaluate the validity of our method in molecule generation, we report several key metrics that capture different aspects of molecule quality and diversity in Table 3 on page 22.

Table 3: Comparison of the generated molecules of pre-trained GDSS model and SVDD applied on various metrics.

| Method | Valid | Unique | Novelty | FCD | SNN | Frag/Test | Scaf | NSPDK MMD | Mol Stable | Atm Stable |
|---|---|---|---|---|---|---|---|---|---|---|
| Pre-trained | **1.0** | **1.0** | **1.0** | 22.2799 | 0.2992 | **0.8274** | 0.0033 | **0.0260** | 0.2903 | **0.9256** |
| SVDD | **1.0** | 0.9375 | **1.0** | **21.5671** | **0.3441** | 0.7803 | **0.0838** | 0.0772 | **0.4783** | 0.9095 |

The validity of a molecule indicates its adherence to chemical rules, defined by whether it can be successfully converted to SMILES strings by RDKit. Uniqueness refers to the proportion of generated molecules that are distinct by SMILES string. Novelty measures the percentage of the generated molecules that are not present in the training set. Fréchet ChemNet Distance (FCD) measures the similarity between the generated molecules and the test set. The Similarity to Nearest Neighbors (SNN) metric evaluates how similar the generated molecules are to their nearest neighbors in the test set. Fragment similarity measures the similarity of molecular fragments between generated molecules and the test set. Scaffold similarity assesses the resemblance of the molecular scaffolds in the generated set to those in the test set. The neighborhood subgraph pairwise distance kernel Maximum Mean Discrepancy (NSPDK MMD) quantifies the difference in the distribution of graph substructures between generated molecules and the test set considering node and edge features. Atom stability measures the percentage of atoms with correct bond valencies. Molecule stability measures

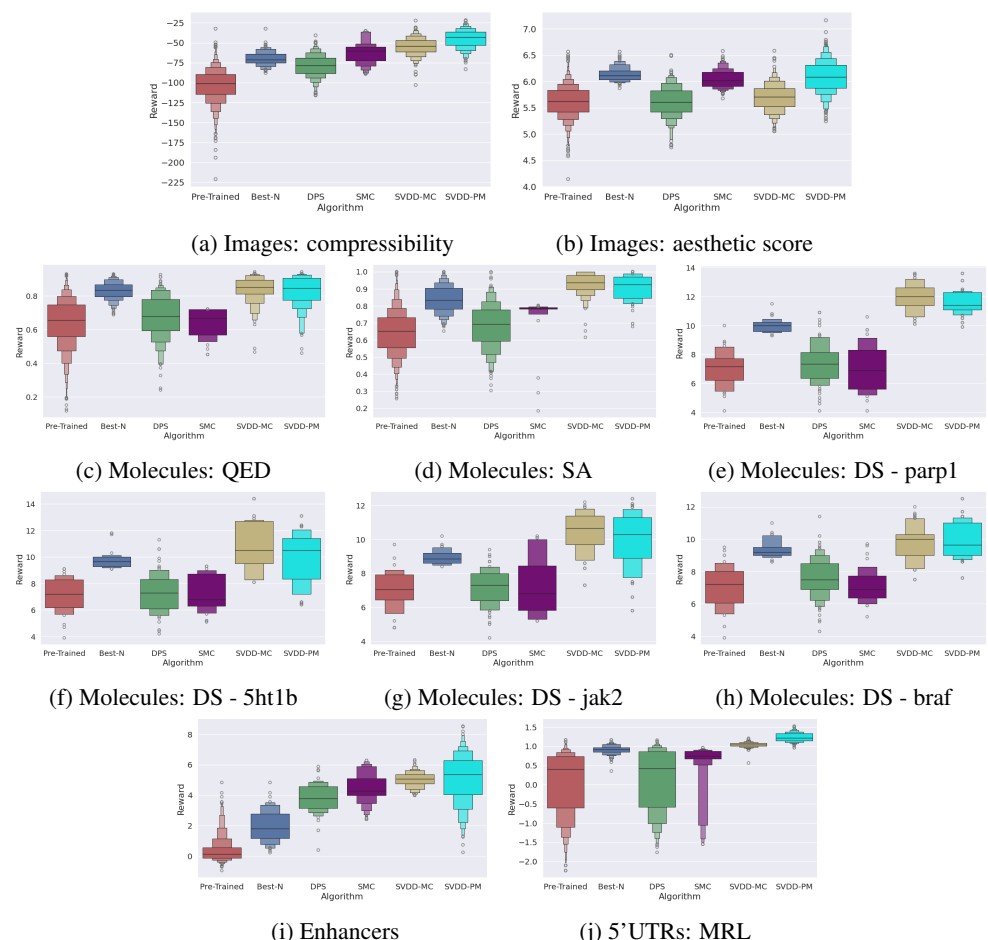

(a) Images: compressibility  (b) Images: aesthetic score

(c) Molecules: QED  (d) Molecules: SA  (e) Molecules: DS - parp1

(f) Molecules: DS - 5ht1b  (g) Molecules: DS - jak2  (h) Molecules: DS - braf

(i) Enhancers  (j) 5'UTRs: MRL

Figure 5: We show the histogram of generated samples in terms of reward functions. We consistently observe that **SVDD** demonstrates strong performances.

the fraction of generated molecules that are chemically stable, *i.e.*, whose all atoms have correct bond valencies. Specifically, atom and molecule stability are calculated using conformers generated by RDKit and optimized with UFF (Universal Force Field) and MMFF (Merck Molecular Force Field).

We compare the metrics using 512 molecules generated from the pre-trained GDSS model and from SVDD optimizing SA, as shown in Table 3. Overall, our method achieves comparable performances with the pre-trained model on all metrics, maintaining high validity, novelty, and uniqueness while outperforming on several metrics such as molecule stability, FCD, SNN, and scaffold similarity. These results indicate that our approach can generate a diverse set of novel molecules that are chemically plausible and relevant.

**More Ablation Studies.** We provide several more ablation studies regarding $M$ on top of the results in the main text, as plotted in Figure 7. The results are consistent with what we have observed in Figure 3.

**Visualization of more generated samples.** We provide additional generated samples in this section. Figure 8 and Figure 9 show comparisons of generated images from baseline methods and **SVDD** with different M values regarding compressibility and aesthetic score, respectively. Figure 10 and Figure 11 presents the comparisons of visualized molecules generated from the baseline model and **SVDD** regarding QED and SA, respectively. The visualizations validate the strong performances of **SVDD**. Given that **SVDD** can achieve optimal SA for many molecules, we also visualize some molecules with optimal SA generated by **SVDD**, as shown in Figure 12. In Figure 13, Figure 14, Figure 15, and Figure 16 we visualizes the docking of **SVDD**-generated molecular ligands to proteins parp1, 5ht1b,

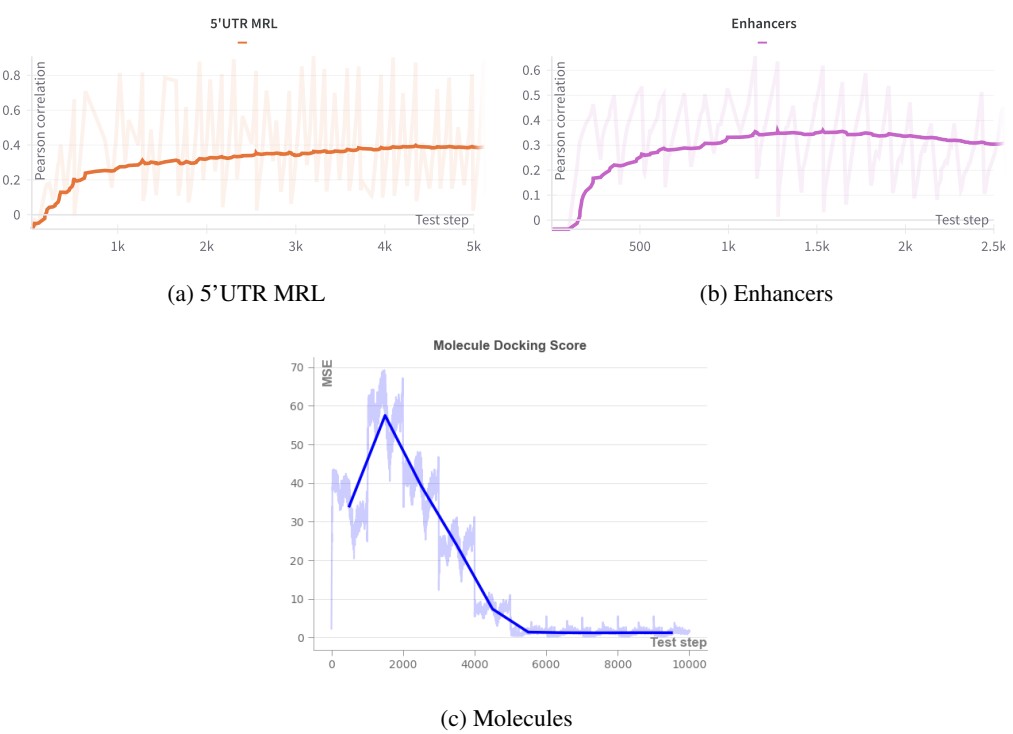

(a) 5'UTR MRL

(b) Enhancers

(c) Molecules

Figure 6: Training curve of value functions

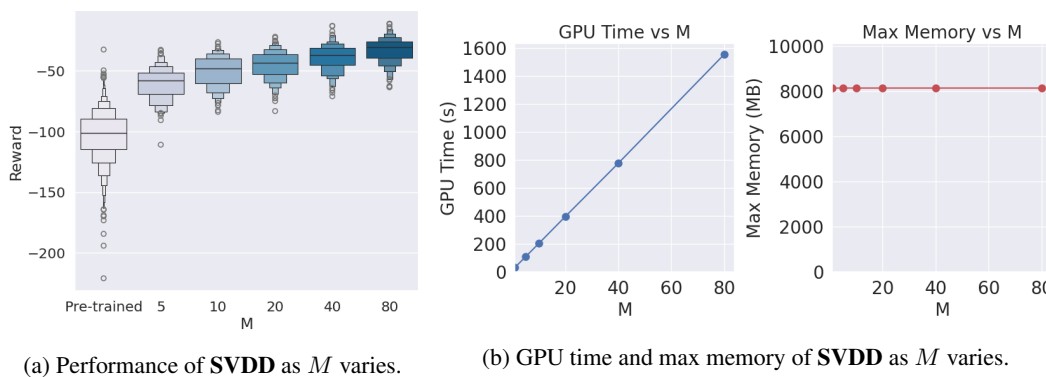

(a) Performance of **SVDD** as $M$ varies.

(b) GPU time and max memory of **SVDD** as $M$ varies.

Figure 7: Abltation Studies (for image generation while optimizing the compressibility)

jak2, and braf, respectively. Docking scores presented above each column quantify the binding affinity of the ligand-protein interaction, while the figures include various representations and perspectives of the ligand-protein complexes. We aim to provide a complete picture of how each ligand is situated within both the local binding environment and the larger structural framework of the protein. First rows show close-up views of the ligand bound to the protein surface, displaying the topography and electrostatic properties of the protein's binding pocket and providing insight into the complementarity between the ligand and the pocket's surface. Second rows display distant views of the protein using the surface representation, offering a broader perspective on the ligand's spatial orientation within the global protein structure. Third rows provide close-up views of the ligand interaction using a ribbon diagram, which represents the protein's secondary structure, such as alpha-helices and beta-sheets, to highlight the specific regions of the protein involved in binding. Fourth rows show distant views of the entire protein structure in ribbon diagram, with ligands displayed within the context of the protein's full tertiary structure. Ligands generally fit snugly within the protein pocket, as evidenced

by the close-up views in both the surface and ribbon diagrams, which show minimal steric clashes and strong surface complementarity.

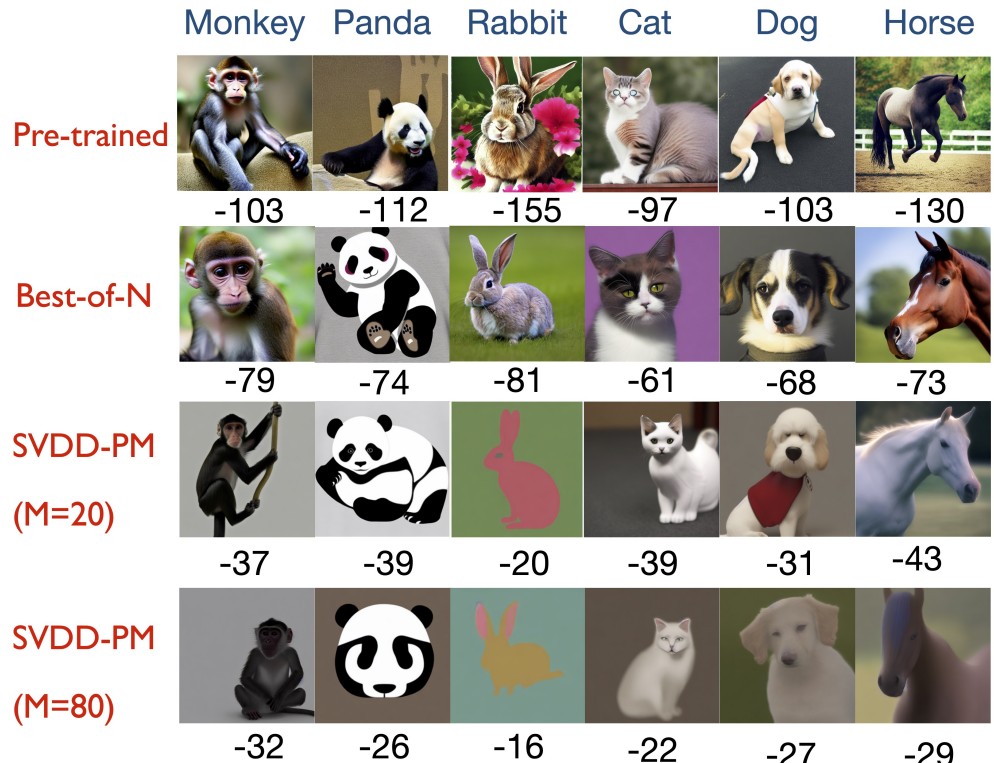

Figure 8: Additional generated samples (Domain: images, Reward: Compressibility)

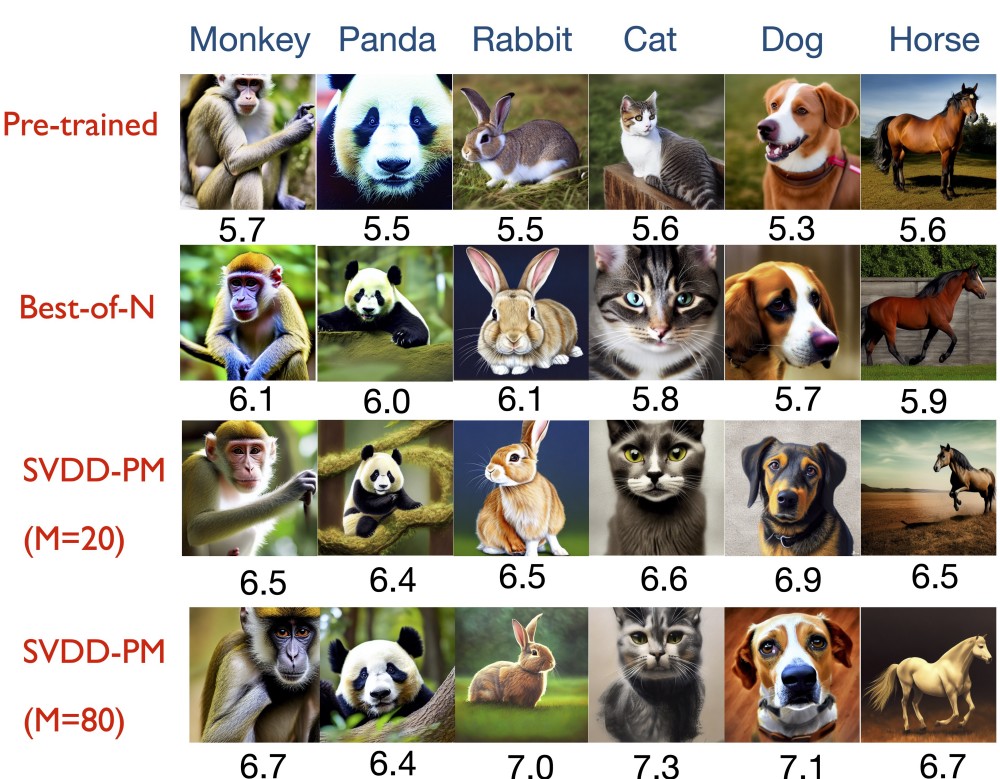

Figure 9: Additional generated samples (Domain: Images, Reward: Aesthetic score)

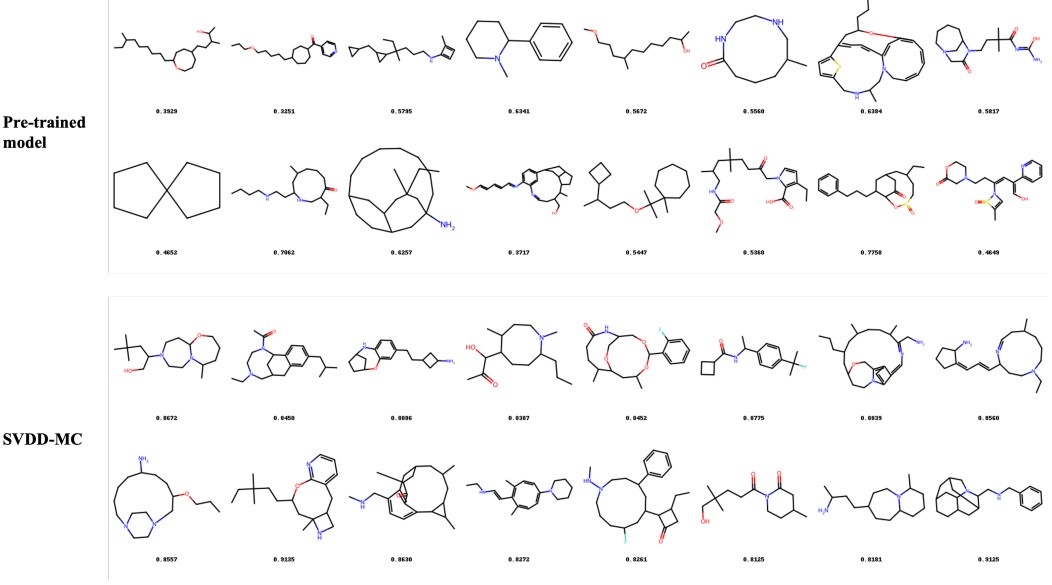

Figure 10: Additional generated samples (Domain: Molecules, Reward: QED score)

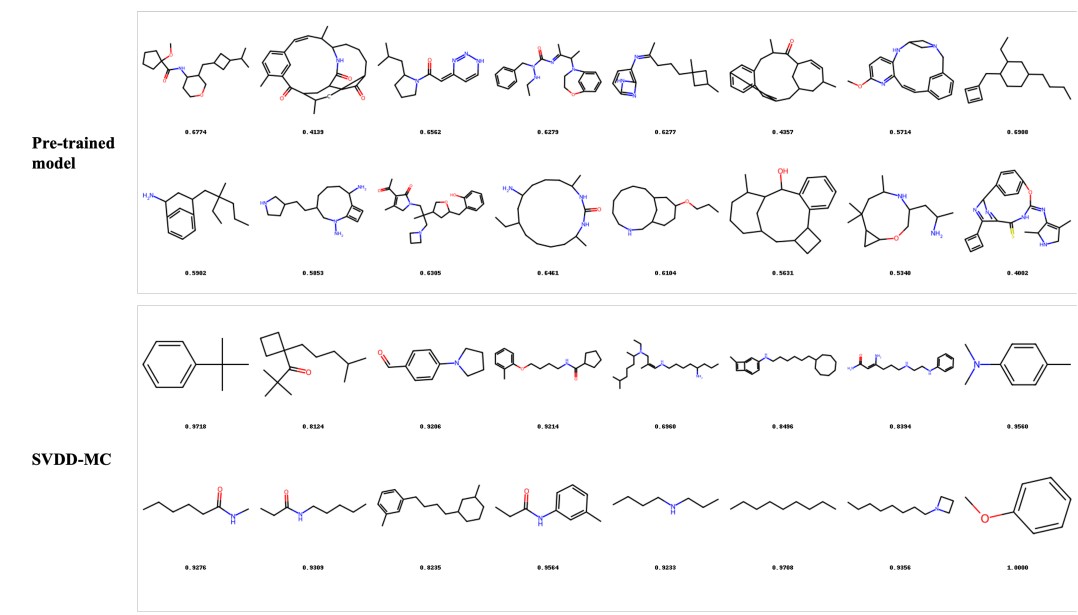

Figure 11: Additional generated samples (Domain: Molecules, Reward: SA score, normalized as $(10 - SA)/9$)

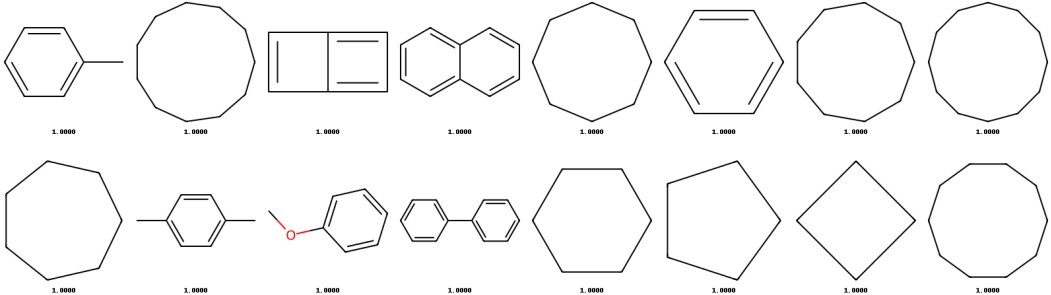

Figure 12: Additional generated samples from **SVDD**-MC (Domain: Molecules, Reward: SA score = 1.0 (normalized as $(10 - SA)/9$))

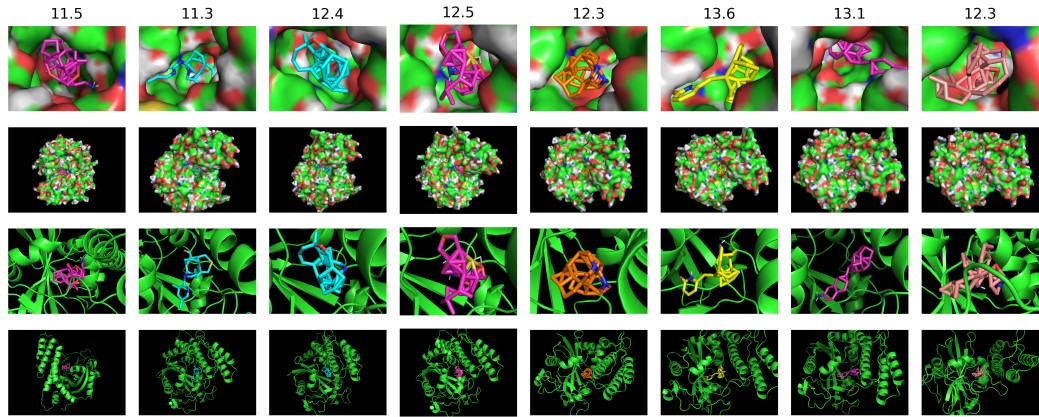

Figure 13: Additional generated samples from **SVDD** (Domain: Molecules, Reward: Docking score - parp1 (normalized as $max(-DS, 0)$))

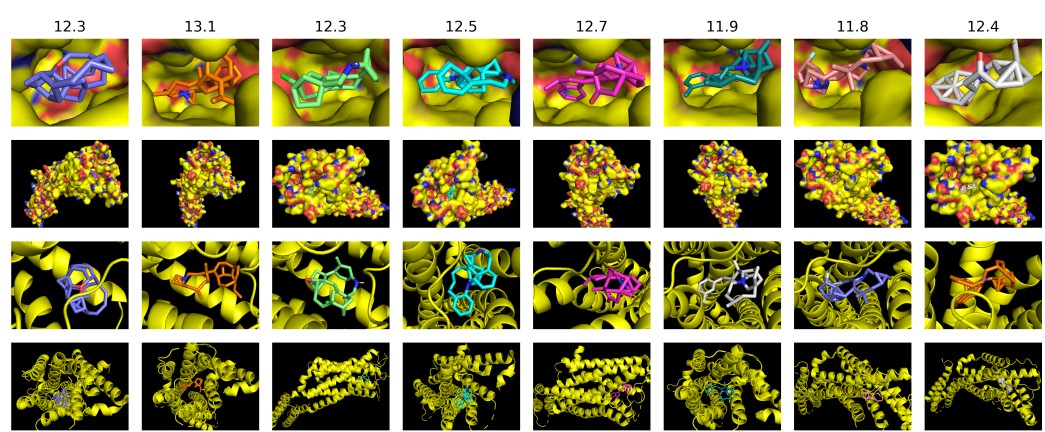

Figure 14: Additional generated samples from **SVDD** (Domain: Molecules, Reward: Docking score - 5ht1b (normalized as $max(-DS, 0)$))

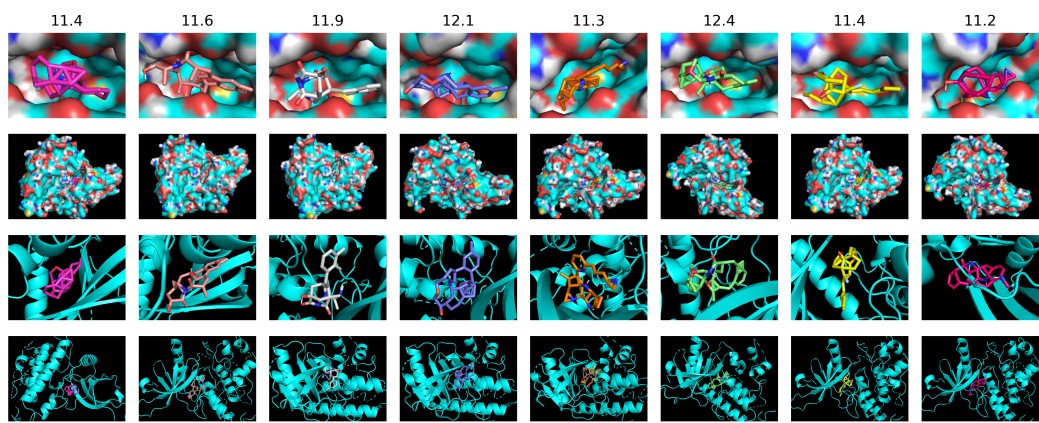

Figure 15: Additional generated samples from **SVDD** (Domain: Molecules, Reward: Docking score - jak2 (normalized as $max(-DS, 0)$))

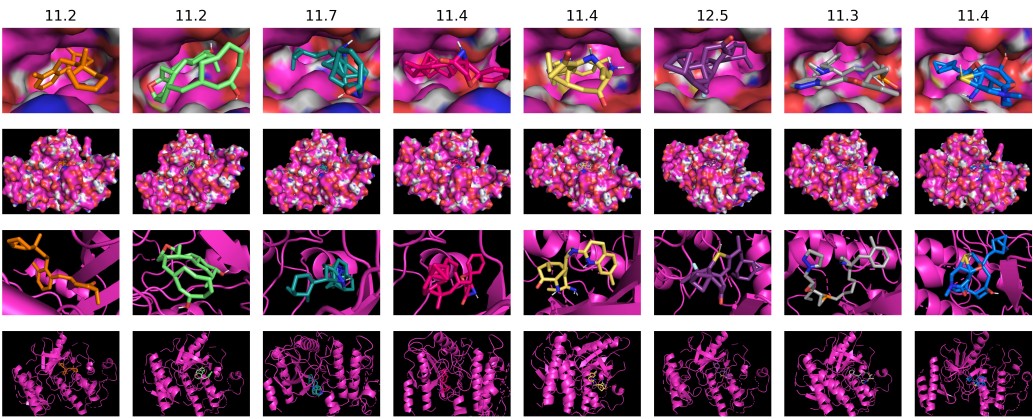

Figure 16: Additional generated samples from **SVDD** (Domain: Molecules, Reward: Docking score - braf (normalized as $max(-DS, 0)$))

