The relative performance of our two proposed methods (**SVDD**-MC or **SVDD**-PM) appears to be domain-dependent. Generally, **SVDD**-PM may be more robust since it does not require additional learning (*i.e.*, it directly utilizes reward feedback). The performance of **SVDD**-MC depends on the success of value function learning discussed in Section F.

**Ablation studies in terms of the duplication size $M$.** We plot the performance of **SVDD**-PM,(when calculating value functions in Line 3 in a non-parallel manner) along with the computational and memory complexity as $M$ varies. First, the performance gradually plateaus as $M$ increases, as shown in Figure 3a. This trend is consistent across all domains. Second, regarding computational complexity, as observed in Figure 3d, it increases linearly with $M$, while memory complexity remains nearly constant. This behavior is expected, as previously mentioned when

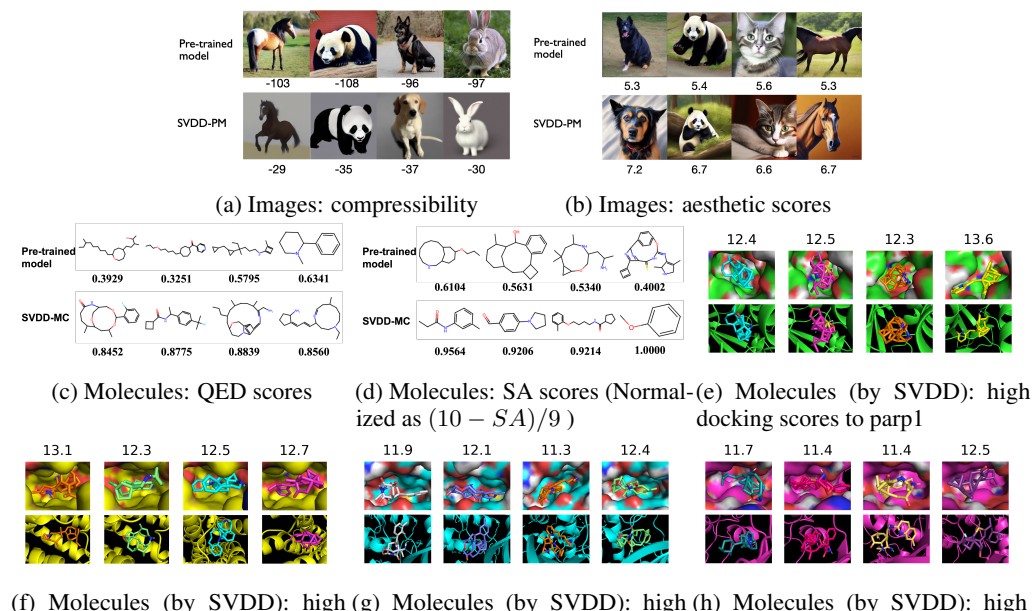

(a) Images: compressibility (b) Images: aesthetic scores

(c) Molecules: QED scores (d) Molecules: SA scores (Normal-(e) Molecules (by SVDD): high ized as $(10 - SA)/9$ ) docking scores to parp1

(f) Molecules (by SVDD): high (g) Molecules (by SVDD): high (h) Molecules (by SVDD): high docking scores to 5ht1b docking scores to jak2 docking scores to braf

Figure 2: Generated samples from our proposal. For more samples, refer to Section F.3. Note that objects in green in (e) or yellow in (f) correspond to the target proteins. The generated small molecules are shown