# OpenReview forum: "Derivative-Free Guidance in Continuous and Discrete Diffusion Models with Soft Value-Based Decoding"
_ICLR.cc/2025/Conference — Submitted to ICLR 2025_

### Official Review · Reviewer_DiPS · 2024-10-30

**Soundness:** 3
**Presentation:** 2
**Contribution:** 2
**Rating:** 3
**Confidence:** 4

**Summary:**

This paper introduces a new method called Soft Value-based Decoding in Diffusion models (SVDD) for optimizing diffusion models to generate samples with desired properties while maintaining naturalness. The contributions include:
- SVDD is an inference-time technique that doesn't require fine-tuning the original diffusion model
- Can work with non-differentiable reward functions, unlike previous methods that required differentiable proxy models
- Applicable to both continuous and discrete diffusion models in a unified way

The algorithm works in the following way:
1. Uses "soft value functions" that predict future rewards from intermediate noisy states
2. At each denoising step:
   - Generates multiple samples using the pre-trained model
   - Selects samples based on their predicted value
There are two variants:
   - SVDD-MC: Uses Monte Carlo regression to learn value functions
   - SVDD-PM: Directly uses reward feedback without additional training

Experimental Results span across multiple domains: image generation, molecule generation, DNA/RNA sequence generation. The proposed method consistently outperformed baseline methods while maintaining sample validity.
The paper demonstrates that SVDD provides an effective way to guide diffusion models toward desired properties while preserving the natural characteristics learned during pre-training.

**Strengths:**

There are several advantages of SVDD over previous methods:
1. No need for differentiable proxy models
2. No fine-tuning required
3. Works with both continuous and discrete spaces
5. Maintains better sample diversity compared to other approaches

The writing of this paper is clear.

**Weaknesses:**

The main weakness is about novelty. To be more specific, I can not see significant difference with twisted SMC methods (e.g., the papers mentioned in Sec 6 and App B). In the writing I see two differences claimed by the authors:
1. In previous works such as Wu et al., the reward is a classifier; while here it is "reward maximization" setting.

First, I think the setting in this work should not be called "reward maximization" but be called "alignment" or "reward sampling" or similar names, due to the reasons in Sec 3.2 "HIGH REWARDS WHILE PRESERVING NATURALNESS". Second, whether the reward function is a classifier is not critical, as even for an usual reward r(x), we can understand it as an unnormalized probability prob(optimality | x).

2. "SMC methods involve resampling across the “entire” batch, which complicates parallelization. Additionally, when batch sizes are small, as is often the case with recent large diffusion model"

I do not quite understand this part. I may miss the difference between SVDD and twisted SMC methods. Does the batch size mean the number of particles in SMC? It will be good if there could be a clarification.

**Questions:**

I do not have further questions.

---

> ### Author Response · Authors · 2024-11-28
>
> We appreciate the reviewer’s feedback on novelty and the comparison with twisted SMC methods. We address you once by explaining that  SVDD and SMC are significantly different, and the performance of SVDD is better than that of the SMC method in the alignment setting, as we wrote in Section 6, 7 And Appendix B. We are happy to answer more if the author still has concerns.
>
> **Q1: Whether the reward function is a classifier is not critical, as even for a usual reward r(x), we can understand it as an unnormalized probability....**
>
> A. Yes, we are on the same page. Indeed, we wrote explicitly in Section 6:
>
> *“They (Wu et al.) can also be applied to reward maximization. Notably, similar to our work, these methods do not require differentiable models”.*
>
> This means that as you said, we did not intend to claim whether we use classifiers or regressors is the primary difference. We indeed show how to do this conversion, i.e., use SMC for reward maximization (or alignment) in Appendix B.1.
>
> However, our main point lies in more different aspects. Here are the primary differences (summary of Appendix B.2.)
>
> * **SVDD is tailored to reward optimization:  SVDD is considered more suitable for optimization than SMC, as empirically shown in Section 7**. This is because, when using SMC for reward maximization, we must set $\alpha$ very low, leading to a lack of diversity. This is expected, as when $\alpha$ approaches 0, the effective sample size reduces to 1. This effect is also evident in our image experiments, as shown in Figure 4. Although SMC performs well in terms of reward functions, there is a significant loss of diversity. Some readers might think this could be mitigated by calculating the effective sample size based on weights (i.e., value functions) and resampling when the effective size decreases; however, this is not the case, as the effective sample size does not directly translate into greater diversity in the generated samples. In contrast, SVDD, maintains much higher diversity.
>
> * **Ease of parallelization in SVDD**: SVDD is significantly easier to parallelize across multiple nodes. In contrast, SMC requires interaction between nodes for parallelization. This advantage is also well-documented in the context of nested-IS SMC versus standard SMC (Naesseth et al., 2019).  More specifically,
>
>        1. Twisted SMC methods typically resample particles across the entire batch, which can introduce sequential dependencies and complicate parallelization on modern hardware (e.g., GPUs). This is because particle resampling requires centralized coordination to determine which particles are retained or replaced.
>
>         2. In contrast, SVDD avoids explicit resampling across the batch by leveraging soft value functions that guide particle updates, enabling fully parallel processing without centralized coordination. This makes SVDD inherently more scalable, particularly for applications involving large diffusion models.
>
> * In SMC, the ``ratio'' is approximated: In SVDD, we approximate each
> $\exp(v_{t-1}(x_{t-1})/\alpha)$
> as a weight. However, in standard SMC, the ratio is approximated as a weight:
> \begin{align*}
>  \frac{\exp(v_{t-1}(x_{t-1})/\alpha) }{\exp(v_{t}(x_{t})/\alpha)}.
> \end{align*}
> The key difference is that in SMC, both the numerator and the denominator are approximated, which could lead to greater error propagation.
>
> **Q. I think the setting in this work should not be called "reward maximization" but be called "alignment" or "reward sampling" or similar names,**
>
> A. We agree that the term "reward maximization" could potentially be interpreted as misleading and appreciate the reviewer’s suggestion to use alternative terminology. However, we avoid using alignment because, in many biology settings, we often use alignment in different ways, such as sequence alignment.
>
> **Q  "SMC methods involve resampling across the “entire” batch, which complicates parallelization. Additionally, when batch sizes are small, as is often the case with recent large diffusion" is unclear.  Does the batch size mean the number of particles in SMC?**
>
> We appreciate the reviewer’s question on batch size and its role in SMC methods. To clarify:
>
> - Yes. In SMC, **batch size refers to the number of particles** (or samples) used in the resampling process as it is more explicitly detailed in Appendix B.2.
> - In twisted SMC methods, resampling operates over **the entire batch**, which involves reweighting and redistributing particles globally. This is computationally expensive and less parallelizable, especially for small batch sizes.
> - In our method (SVDD), batch size refers to the number of samples evaluated in parallel during diffusion, and resampling is replaced by value-based guidance. This avoids the bottleneck of centralized resampling and ensures scalability even with smaller batch sizes.
>
> We will expand on these points in Section 6 to improve clarity and explicitly highlight the differences between SVDD and twisted SMC methods.

---

### Official Review · Reviewer_Jmxu · 2024-11-02

**Soundness:** 2
**Presentation:** 2
**Contribution:** 1
**Rating:** 3
**Confidence:** 5

**Summary:**

This paper introduces SVDD which is a new method to fine-tune diffusion models in both continuous and discrete spaces. SVDD-MC learns a soft value function by regression to $x_t$ while SVDD-PM exploits the posterior mean parametrization of masked diffusion models to estimate the soft value function directly. Given a soft value function, SVDD can be applied to a general class of reward functions, including non-differentiable ones, at inference time without further fine-tuning. This can be seen as a variation of the famous Sequential Monte-Carlo algorithm but applied and modified for diffusion models. Experiments are done on images, molecules, and docking and show improvements in fine-tuning performance under a specified reward metric.

**Strengths:**

The paper tackles a timely problem in considering fine-tuning diffusion models. Moreover, the suggested approach of SVDD-PM enjoys being computationally cheap to use as it does not require any further training while both SVDD-MC and SVDD-PM are applicable in settings where the reward function is non-differentiable. This is impactful because this unlocks a lot of potential application domains that have black-box rewards where learning a surrogate reward model is non-trivial. Finally, the paper considers a diverse set of experimental settings to showcase the universality of the proposed approach.

**Weaknesses:**

While the paper has some notable strengths there are a few lingering questions that point to potential weaknesses. I will try to list them below.

**Theoretical weaknesses**

Two main questions arise when looking at the setup. The first one is the actual target distribution and whether SVDD hits it. In the continuous setting, I have severe doubts about whether the correct terminal distribution is reached due to the initial value function bias problem as introduced in Adjoint Matching (Domingo-Enrich et. al 2024). Certainly, nothing in the current theory suggests the process during fine-tuning is memoryless. Moreover, it is unclear what ramifications and bias we introduce in the MC setup when the regression is done using $r(x_0) \to x_t$ as opposed to the more numerically unstable soft value function. For example, I believe when you remove the $\exp$ from your regression target, this is a based value function but there is no discussion on this point outside of the fact that it is less numerically stable. As a result, I am dubious about the claims made about hitting the correct target distribution.


Another question is the connection to Sequential Monte Carlo. There is a discussion on this in the paper but I think it's not accurate enough. I disagree with the statement made in the paper. The algorithm you propose is quite literally SMC but adapted to reward maximization, there is even a resampling step which is exactly what is done in SMC. The arguments that SMC is done over a batch are lukewarm. There is nothing wrong with demonstrating that SMC can be effectively applied to sampling from discrete diffusion---like analogously done for an autoregressive model by Zhao et. al 2024---and this is a valuable contribution. I suggest the authors be a bit more forthright with their claims as I would buy it a lot more. In fact, with the right framing, you achieve novelty by showing how SMC applies to a newer more interesting problem domain.

**Additional Technical weaknesses**

One of the main selling points of SVDD is the fact that it is supposed to be a cheap inference time algorithm. This I believe is not quite true because of the need to estimate the soft value function in SVDD-MC. Indeed, one must estimate the soft value function using rollouts which I believe adds a heavy pre-processing step. I also did not see SVDD-MC in the ablation studies about computational cost---likely because it's significantly more expensive than SVDD-PM. Thus, I believe the main claim for SVDD-MC being a lightweight method is a bit misleading. Of course, if you had the perfect estimated value function then inference scales as indicated in the plot for 3c,d but this is not the full picture.

**Experimental weaknesses**

A glaring missing baseline is Relative Trajectory Balance (Venkataraman et. al 2024) which does fine-tuning exactly like this paper considers for both discrete and continuous diffusion models. I kindly request the authors to consider adding this important baseline. Moreover, it is a bit surprising that there is no text experiment given the heavy emphasis on using Masked Diffusion Models which have primarily been introduced for text. I would be encouraged to see a text experiment---perhaps of a similar scale to Zhao et. al 2024---to highlight that SVDD can be applied in the text setting.

The current experimental findings in Table 2 are not complete as they do not show other important aspects of the generated sample. They simply show that reward is maximized but this could also happen through gamification of the reward function. For instance, I would appreciate the authors providing sample-based diversity metrics to quantify how bad the drop in diversity is among the baselines. At the very minimum, FID scores for images should be provided and I'll let the authors determine appropriate diversity metrics for the other domains to complement the findings in Table 2.

**Closing remarks**

Having said all of these weaknesses, I will note that I am open to significantly raising my score if **all of my concerns** are adequately addressed to my level of satisfaction. I will also state that I did not read the appendix so if I have missed something I would appreciate a pointer to the result there.

I encourage the authors in their rebuttal endeavors and I hope they can strengthen the paper which I would like to eventually recommend for acceptance but not in its current state.


**References**

Venkatraman, Siddarth, et al. "Amortizing intractable inference in diffusion models for vision, language, and control." arXiv preprint arXiv:2405.20971 (2024).

Zhao, Stephen, et al. "Probabilistic inference in language models via twisted sequential monte carlo." arXiv preprint arXiv:2404.17546 (2024).

**Questions:**

I would appreciate it if the authors could address my theoretical concerns regarding 1.) what is the optimal distribution hit by SVDD 2.) what is the bias introduced in the MC estimate and 3.) the actual computational cost of SVDD-MC.

In addition, I would appreciate it if the authors could carry out the additional experiments I have suggested with added diversity quantification.

---

> ### Author Response · Authors · 2024-11-30
>
> We appreciate your detailed review. We have addressed your concerns by explaining (1) an initial bias problem is already considered, (2) more details on the approximation error of SVDD-MC, (3) the difference between SMC and SVDD, (4) why we don’t emphasize diversity metrics and comparison with fine-tuning methods like RTB.
>
> > *W. Does an initial bias problem exist? Relation with Domingo-Enrich et al. 2024*
>
> **There is no contraction between our paper and Domingo-Enrich et al. 2024, and our Theorem 1 considers an initial bias problem.** Since Domingo-Enrich et al. 2024 use continuous-time formulation, but we use discrete-time formulation, it might lead to confusion. However, our statement says we need to sample from the exponentially weighted initial distribution but not the original one, which is consistent with Domingo-Enrich et al. 2024. While their work deal with it by changing the schedule but without changing initial distributions, we handle it by sampling from a weighted initial distribution with value functions.
>
> > *W. Approximation error in SVDD-MC*
>
> We acknowledge that our approximation is heuristic, and we will add more discussion. However, we emphasize that in SVDD-MC, our algorithm works well without this heuristic. **We claim that empirically, the algorithm works more stable with this heuristic.**
>
> > *W. Difference/Relation with Sequential Monte Carlo.*
>
> **While we don’t intend to claim, our algorithm is not related to SMC (indeed, we exactly claim that our algorithm is known as nested-SMC in the SMC literature in Appendix B.2.), a naively adapted version of Zhao et al. 2024 is different from our algorithm.** That’s why we distinguish between SMC and SVDD. This naively adapted version is specified in Appendix B.1, and this is different from SVDD. We are happy to answer more if certain points are unclear.
>
> Here are the primary differences (summary of Appendix B.2.)
>
> * **SVDD is tailored to reward optimization:  SVDD is considered more suitable for optimization than SMC, as empirically shown in Section 7**. This is because, when using SMC for reward maximization, we must set $\alpha$ very low, leading to a lack of diversity. This is expected, as when $\alpha$ approaches 0, the effective sample size reduces to 1. This effect is also evident in our image experiments, as shown in Figure 4. Although SMC performs well in terms of reward functions, there is a significant loss of diversity. Some readers might think this could be mitigated by calculating the effective sample size based on weights (i.e., value functions) and resampling when the effective size decreases; however, this is not the case, as the effective sample size does not directly translate into greater diversity in the generated samples. In contrast, SVDD, maintains much higher diversity.
>
> * **Ease of parallelization in SVDD**: SVDD is significantly easier to parallelize across multiple nodes. In contrast, SMC requires interaction between nodes for parallelization. This advantage is also well-documented in the context of nested-IS SMC versus standard SMC (Naesseth et al., 2019).  More specifically,
>
>        1. Twisted SMC methods typically resample particles across the entire batch, which can introduce sequential dependencies and complicate parallelization on modern hardware (e.g., GPUs). This is because particle resampling requires centralized coordination to determine which particles are retained or replaced.
>
>         2. In contrast, SVDD avoids explicit resampling across the batch by leveraging soft value functions that guide particle updates, enabling fully parallel processing without centralized coordination. This makes SVDD inherently more scalable, particularly for applications involving large diffusion models.
>
> * In SMC, the ``ratio'' is approximated: In SVDD, we approximate each
> $\exp(v_{t-1}(x_{t-1})/\alpha)$
> as a weight. However, in standard SMC, the ratio is approximated as a weight:
> \begin{align*}
>  \frac{\exp(v_{t-1}(x_{t-1})/\alpha) }{\exp(v_{t}(x_{t})/\alpha)}.
> \end{align*}
> The key difference is that in SMC, both the numerator and the denominator are approximated, which could lead to greater error propagation.
>
> > *W. Computational Cost of SVDD-MC*
>
> A. We acknowledge the reviewer's concern about the computational cost of SVDD-MC. Below, we address this in detail:
>
> We acknowledge that estimating the soft value function in SVDD-MC involves additional rollouts, which introduce computational overhead. However, this computation is minor compared to that needed for fine-tuning the pre-trained diffusion model. Indeed, in classifier guidance, we have a similar situation. **However, to our knowledge, the diffusion model community agrees that training classifiers is much easier than fine-tuning generative models because training classifiers is technically just supervised learning.**
>
> Furthermore, our paper recommends using the SVDD-PM method when it is hard to train in an MC way.

---

> > ### Author Response · Authors · 2024-11-30
> > **Continue**
> >
> > Experimental Weaknesses
> >
> > > A glaring missing baseline is Relative Trajectory Balance (Venkataraman et. al 2024) which does fine-tuning exactly like this paper considers for both discrete and continuous diffusion models. I kindly request the authors to consider adding this important baseline.
> >
> > We thank the reviewer for pointing out the missing baseline. We know that Relative Trajectory Balance (RTB) is a relevant method for fine-tuning in diffusion models. **As we are not intentionally adding any tuning methods (Black et al., 2023; Fan et al., 2023) as well as RTB, we don’t plan to add a comparison.
> >
> > **In our work, we acknowledge that we don’t intend to claim our method like infernece-time technique  is better than fine-tuning methods in Section 5.3.This is because it is hard to consider the right comparison between accounting training time and inference time. Indeed, many representative papers on inference-time techniques like our work do not compare fine-tuning generative models.**
> >
> > [1] Zhao, S., R. Brekelmans, A. Makhzani, and R. Grosse (2024). Probabilistic inference in language models via twisted sequential monte carlo. ICML
> >
> > [2] Chung, H., J. Kim, M. T. Mccann, M. L. Klasky, and J. C. Ye (2023). Diffusion posterior sampling for general noisy inverse problems. ICLR
> >
> > [3] ] Ho, J., T. Salimans, A. Gritsenko, W. Chan, M. Norouzi, and D. J. Fleet (2022). Video diffusion models. Advances in Neural Information Processing Systems 35, 8633–8646.
> >
> > [4]] Bansal, A., H.-M. Chu, A. Schwarzschild, S. Sengupta, M. Goldblum, J. Geiping, and T. Goldstein (2023). Universal guidance for diffusion models. In Proceedings of the IEEE/CVF Conference on Computer Vision and Pattern Recognition,
> >
> >
> > > W. Missing Text Experiment
> >
> > - We appreciate the suggestion to include a text experiment.  Since we alrady have tried in several domains, we defer it fo tufure work.
> >
> > > W. Diversity Metrics
> >
> > First, we have certain discussion in molecular domains. Here, as shown in Table 3, we report diversity and validity metrics, e.g., Validity percentage, Uniqueness percentage, and Novelty percentage. These results demonstrate that SVDD maintains diversity and naturalness comparable to baselines while achieving higher rewards. We will include more naturalness metrics to evaluate sample quality in more domains.
> >
> > In general, although we recognize that higher diversity is better, **representative papers about alignment in diffusion models ([1], [2], [3]), like our work, don’t have diversity metrics. This is because alignment (reward maximization) is a primary goal, and diversity is a secondary objective.** Instead, people often show generated samples to show the diversity we did. As another reason, we believe this is often a subjective metric very suspect to the definition of distance.
> >
> > [1] Clark, K., P. Vicol, K. Swersky, and D. J. Fleet (2024). Directly fine-tuning diffusion models on differentiable rewards. arXiv preprint ICLR
> >
> > [2] Fan, Y., O. Watkins, Y. Du, H. Liu, M. Ryu, C. Boutilier, P. Abbeel, M. Ghavamzadeh, K. Lee, and K. Lee (2023). DPOK: Reinforcement learning for fine-tuning text-to-image diffusion models. arXiv preprint NeurIPS
> >
> > [3] Black, K., M. Janner, Y. Du, I. Kostrikov, and S. Levine (2024). Training diffusion models with reinforcement learning

---

### Official Review · Reviewer_YMgn · 2024-11-02

**Soundness:** 3
**Presentation:** 3
**Contribution:** 3
**Rating:** 5
**Confidence:** 3

**Summary:**

This work provides a unified framework of guidance in diffusion models, both discrete and continuous, with minimal additional training and applicability in domains where a downstream reward might not even be differentiable. The proposed method SVDD (MC and PM) is applicable in discrete diffusion where a continuous gradient of energy cannot be directly added to the discrete state space, as well as cases where the reward is non differentiable which is the case in a lot of scientific domains. The work tackles an important problem in the scientific domain and leads to controllable generation without having to fine-tune large scaled models. Their results show that generations from SVDD lead to higher downstream rewards than the baselines considered.

**Strengths:**

- The authors provide a widely applicable method that can be applied both to discrete and continuous diffusion settings.
- The proposed method, unlike previous guidance algorithms, does not rely on an explicitly trained conditional diffusion model (eg. for classifier free guidance), or on differentiable reward terms (eg. classifier based guidance).
- Results on both image and scientific domains highlight the benefits of the approach towards controlled generation of objects with high downstream reward, as intended.
- The work also conducts experiments in a wide variety of domains, ranging from images, molecules and DNA.

**Weaknesses:**

- The authors consider setting $\alpha=0$ in their experiments. However, prior work highlights that setting $\alpha=0$ leads to over-optimization and increasingly reduces the diversity and realistic nature of the samples. Could the authors provide clarity on why this is not a problem in their setup?
- The work relies on two major assumptions (one for SVDD-MC and the other for SVDD-PM), which are neither well motivated theoretically nor are there any details provided about it.
- **Assumption about SVDD-MC**: The authors replace the logarithm of expectation with the expectation of logarithm in their quantity of interest, which in reality is only a bound on the actual quantity. Could the authors consider experimenting on some synthetic domain to describe the bias and variance caused by this approximation? When is this approximation reasonable and when would it be extremely incorrect?
- **Assumption about SVDD-PM**: This algorithm combines the above approximation with pushing the expectation inside the reward function $r(\cdot)$. As with above, could the authors conduct experiments on synthetic domains and highlight when and where such an assumption is reasonable, and when is it violated?
- While the approach leads to generation of samples with high reward, the authors do not provide any kind of metrics that test for diversity of the samples generated.

**Questions:**

- Is there a typo under the first equation in Section $4.1$, where the expectation is induced by $p_t^{pre}(\cdot | x_{t-1}) -$ note the negative instead of positive sign.

---

> ### Author Response · Authors · 2024-11-28
>
> Thank you very much for your rebuttal. We have addressed your concern by explaining (1) why $\alpha=0$ is fine, (2) approximation errors in value function learning, and (3) how diversity is discussed in the works of alignment in diffusion models.
>
> > W1: The authors consider setting $\alpha=0$  in their experiments. However, prior work highlights that setting leads to over-optimization and increasingly reduces the diversity and realistic nature of the samples. Could the authors provide clarity on why this is not a problem in their setup?**
>
> A. We thank the reviewer for pointing out concerns regarding setting $\alpha=0$.
>
> - **How naturalness (realisticy) is retrained?**: We always sample from pre-trained models because we use them as proposal distributions. Hence, it is naturally expected the likelihood is high.  Indeed, empirically, the images shown in the Figures are realistic. We also have more quantitative metrics in molecules like Table 3.
>
> - **How diversity is retained**: While we recognize its importance, we intend to refrain from claiming our goal is to retain diversity because our primary goal is to optimize rewards while maintaining naturalness.  However, even if $\alpha=0$, it is expected that the diversity is not lost because it is regarded that we are sampling from many modes on the distribution of $p_{pre}(x)\exp(r(x)/\alpha)$ (with small $\alpha$ but not $0$ exactly) due to the randomness coming from pre-trained models and finite $M$ in practice. Indeed, we observe it as we show in molecules and images. Furthermore, as shown in Table 3, we report diversity metrics for molecular domains, e.g., Uniqueness percentage and Novelty percentage.
>
> As far as I know, **representative papers about alignment in diffusion models ([1], [2], [3]) like our work don’t show diversity metrics.** This is because alignment (reward maximization) is a primary goal, and diversity is a secondary objective.  Instead, people often show generated samples to show the diversity we did.
>
> [1] Clark, K., P. Vicol, K. Swersky, and D. J. Fleet (2024). Directly fine-tuning diffusion models on differentiable rewards. arXiv preprint ICLR
>
> [2]  Fan, Y., O. Watkins, Y. Du, H. Liu, M. Ryu, C. Boutilier, P. Abbeel, M. Ghavamzadeh, K. Lee, and K. Lee (2023). DPOK: Reinforcement learning for fine-tuning text-to-image diffusion models. arXiv preprint NeurIPS
>
> [3]  Black, K., M. Janner, Y. Du, I. Kostrikov, and S. Levine (2024). Training diffusion models with reinforcement learning
>
> > W2, 3, &4  Concerns on value function approximation errors of SVDD-MC and SVDD-PM .
>
> A. We appreciate the reviewer’s concerns regarding the assumptions underlying SVDD-MC and SVDD-PM. In general, we admit these approximations are heuristic and we will add more discussion. **However, we want to convey that (1) in SVDD-MC, our algorithm works well without this heuristic, and we just empirically observe the algorithm works more stable with this heuristic; (2) in SVDD-PM, this is a well-known empirically successful approximation in current related literature ([1],[2], [3]) as Remark in 2, such as classifier guidance variants when rewards are classifiers.**
>
> [1] Chung, H., J. Kim, M. T. Mccann, M. L. Klasky, and J. C. Ye (2023). Diffusion posterior sampling for general noisy inverse problems. ICLR
>
> [2] Ho, J., T. Salimans, A. Gritsenko, W. Chan, M. Norouzi, and D. J. Fleet (2022). Video diffusion models. Advances in Neural Information Processing Systems 35, 8633–8646.
>
> [3] Bansal, A., H.-M. Chu, A. Schwarzschild, S. Sengupta, M. Goldblum, J. Geiping, and T. Goldstein (2023). Universal guidance for diffusion models. In Proceedings of the IEEE/CVF Conference on Computer Vision and Pattern Recognition,
>
> > W5 While the approach leads to generation of samples with high reward, the authors do not provide any kind of metrics that test for diversity of the samples generated.
>
> A. Discussed in W1.
>
> > Q1 Potential typo
>
> A. We appreciate the reviewer catching this potential typo. The negative sign in the expectation induced by p_t^{\text{pre}}(\cdot | x_{t-1}) is not a typo. The sign appears due to the reward gradient term, which accounts for minimizing divergence from the pre-trained prior while maximizing the reward. We will clarify this explicitly in the revised manuscript to avoid confusion.

---

### Official Review · Reviewer_v5um · 2024-11-04

**Soundness:** 1
**Presentation:** 2
**Contribution:** 2
**Rating:** 3
**Confidence:** 4

**Summary:**

The paper introduces SVDD, a method which aims to sample from the product distribution $p^*(x_0) \propto p^{pre}_0(x_0) \exp(R(x_0)/\alpha)$ for some non-negative reward function $R(x)$, constant $\alpha \geq 0$ and pre-trained diffusion model $p^{pre}_t(x_t | x_{t + 1})$.

The method employs an SMC-inspired procedure for iteratively sampling and reweighting samples according to a soft value function and its corresponding optimal policy.  Providing two options to obtain the soft-value function (which is required for the method's importance sampling step), the authors show that SVDD can be used with a cheap approximation based on the diffusion model's denoiser or an amortized version based on regression to a Monte Carlo estimate.  The authors evaluate the performance of SVDD on a series of tasks -- images, molecule design, and DNA/RNA design.

**Strengths:**

The paper presents a number of strengths, such as

- Presenting a novel application of nested Sequential Monte Carlo to the difficult problem of sampling from the product distribution $p^*(x_0)$ given a pre-trained diffusion model.
- The method, especially SVDD-PM, provides a particularly efficient method to sample from the product distribution when no differentiable reward is available and the reward function is cheap.  The manuscript shows that SVDD can indeed increase the reward over the pre-trained model, offering a compelling option to sample from the target product distribution with little overhead.
- The problem of cheaply sampling from the product distribution in the presence of non-differentiable rewards is especially significant as existing methods typically require availability of gradients or expensive (typically simulation-based) fine-tuning algorithms.  Non-differentiable rewards are often seen in scientific discovery, a target area aptly pointed out by the authors.

**Weaknesses:**

Overall I had a some issues regarding clarity of the paper, concerns about sources of bias that are not discussed in the manuscript, and am worried that the experimental section does not paint a fair picture of SVDD's performance relative to baselines.  I will discuss each of these in turn



### Unclear focus of probabilistic inference vs reward maximization

Section 3.2 states that the objective of this paper is to perform probabilistic inference and sample from the target distribution $p^*(x_0) \propto p^{pre}(x_0)\exp(R(x_0) / \alpha)$.  However, towards the beginning of the experiment section and throughout the appendix the manuscript begins to say that SVDD is actually meant for reward maximization, not the problem of sampling from $p^*(x_0)$.  In particular, the manuscript states that in practice they set $\alpha = 0$ for all experiments, which corresponds to a constrained reward maximization where $p^*(x_0)$ is a Dirac centered at $x_0^* = \underset{x_0 \in Support(p_0^{pre}(x_0))}{\arg\max}R(x_0)$.  This is quite different from sampling from the $p^*(x_0)$ for any $\alpha$ and if this is the goal of SVDD it should be compared to baselines which try to do reward maximization.



### Missing discussion and investigation on bias of soft value function estimates

The manuscript defines the soft value function as $v_t(x_t) = \alpha \log \mathbb{E}_{x_0 \sim p^{pre}(x_0 | x_t)}[\exp(R(x_0) / \alpha)]$.  Next, due to issues with numerical stability for small values of $\alpha$ the authors make use of an approximation

$v_t(x_t) = \alpha \log \mathbb{E}_{x_0 \sim p^{pre}(x_0 | x_t)}[\exp(R(x_0) / \alpha)]$

$ \approx \alpha \log \exp(\mathbb{E}_{x_0 \sim p^{pre}(x_0 | x_t)}[R(x_0)] / \alpha)$

$ = \mathbb{E}_{x_0 \sim p^{pre}(x_0 | x_t)}[R(x_0)]$

The second step takes the $\exp$ function outside of the expectation and as such requires an application of Jensen's inequality, implying that $v_t(x_t) \geq \mathbb{E}_{x_0 \sim p^{pre}(x_0 | x_t)}[R(x_0)]$.  This means that the Monte Carlo regression used for SVDD-MC is in fact biased (although consistent), a fact which is not mentioned in the paper.

The situation is more complicated for SVDD-PM which first applies Jensen's and then another approximation as

$v_t(x_t) \geq \mathbb{E}_{x_0 \sim p^{pre}(x_0 | x_t)}[R(x_0)]$

$ \approx R(\mathbb{E}_{x_0 \sim p^{pre}(x_0 | x_t)}[x_0])$

It is unclear to me whether the error of the posterior mean estimate can be shown to be bounded as the reward function is potentially non-convex, but would be happy if the authors had some insight into this.

Given that SVDD requires accurate estimates of the soft-value functions to sample from the target distribution $p^*(x_0)$ I would be more convinced of SVDD's abilities were there a more detailed (potentially including an empirical results) analysis of the bias of the Monte Carlo regression and posterior mean estimates.



### Issues with inconsistent setting of $\alpha$ for baselines

The stated goal of SVDD is to sample from the target distribution $p^*(x_0;\alpha) \propto p^{pre}(x_0)\exp(R(x_0) / \alpha)$, where the temperature parameter $\alpha$ controls how peaky $p^*(x_0;\alpha)$ becomes.  As discussed above, as $\alpha \rightarrow 0$ the target distribution becomes focused on the maximizer of the reward which is in the support of the pretrained model such that

$\mathbb{E}_{x \sim p^* (x;0^+)}[R(x)] = \underset{x_0 \in Support(p^{pre}(x_0))}{\arg\max}R(x_0)$.

In general, as the value of $\alpha$ is decreased the expected reward under the target distribution should increase.  As such, comparing the distribution of rewards of generated samples for methods using different values of $\alpha$ does not paint an accurate picture of each method's performance as one method having a higher quantile reward may simply be a consequence of the setting of $\alpha$.

Unfortunately, the manuscript's experiments use significantly different values of $\alpha$ for its method and baselines while using the reward at different quantiles as the main performance metric.  This is more problematic as the value of $\alpha$ for their method is set to $0$ (where the true expected reward is the maximum reward value) and a larger value of $\alpha$ for baselines.  Because the value of $\alpha$ is not set to be equal for SVDD and all baselines I do not believe that the experimental results in Table 2 paint a fair picture of SVDD's performance.



### Overall comments

I generally have concerns with the settings of either the number of particles $M$ being too small or the bias of the soft-value function estimates being too large.  As far as I understand (and perhaps I am missing something!) by setting $\alpha=0$ for SVDD in the experiment section the method _should_ be suffering from mode collapse and generating very high reward samples as the target distribution $p^*(x_0)$ is a Dirac centered at $x_0^* = \underset{x \in Support(p_0^{pre}(x))}{\arg\max}R(x)$.  However, samples from SVDD do not exhibit this expected mode collapse, which seems to indicate that either many more particles $M$ need to be used or the bias from the value function estimation is preventing the algorithm from properly sampling from the target distribution.

I note that the main reason for my score mostly due to the issue with inconsistent setting of $\alpha$ for SVDD and baselines in the experiments section as well as the missing discussion.  A missing discussion/analysis of the bias of the value function estimates and their impact on SVDD's ability to sample from the target distribution also contributes significantly to my score.

**Questions:**

1. I did not see it mentioned in the manuscript -- how many seeds were used for experiments?
2. Could the authors provide a discussion of the relation between the consistency of nested SMC and the consequence of using more or less particles in their method?
3. How expensive is training the soft-value function estimate in SVDD-MC?  If it is reasonably long would it be worth adding a fine-tuning based method (e.g., relative trajectory balance https://arxiv.org/abs/2405.20971).  On the other hand, if training the soft-value estimate is especially cheap it would be worthwhile to emphasize this more in the manuscript as a benefit of this method compared to direct fine-tuning methods.
4. Since the goal of SVDD is to sample from the product distribution of reward and pretrained model could they add some metrics evaluating the diversity of their samples or their naturalness (e.g., likelihood under the pre-trained model when available)?

---

> ### Author Response · Authors · 2024-11-29
>
> We appreciate your detailed review. We have clarified (1) our reward maximization goal and choice of $\alpha$ and (2) approximation errors when learning value functions.
>
> > W1. Unclear Focus on Probabilistic Inference vs Reward Maximization
>
> Yes, our focus is reward maximization. **This target distribution is widely accepted in alignment problems, as used in many representative papers in RLHF. While it can be seen as a probabilistic inference, the literature focuses on the reward maximization aspect to our knowledge [1,2,3].** (i.e., do not have metrics on diversity in general). But, we acknowledge we should have added more likelihood metrics. We will cite these papers and clarify the goal.
>
> [1]  Ziegler, Daniel M., et al. "Fine-tuning language models from human preferences." arXiv preprint arXiv:1909.08593 (2019).
>
> [2] Rafailov, R., Sharma, A., Mitchell, E., Manning, C. D., Ermon, S., and Finn, C. Direct preference optimization:Your language model is secretly a reward model. In Thirtyseventh Conference on Neural Information Processing Systems, 2023.
>
> [3] Ethayarajh, Kawin, et al. "Kto: Model alignment as prospect theoretic optimization." (2024).
>
> > W2: Missing discussion and investigation on bias of soft value function estimates
>
> We appreciate the reviewer’s concerns regarding the assumptions underlying SVDD-MC and SVDD-PM. In general, we admit these approximations are heuristic and we will add more discussion.
>
> - Approximation: SVDD-MC, our algorithm works well without this heuristic, but we just say that empirically, the algorithm works more stable with this heuristic. We plan to add more to the discussion.
>
> - Approximation of SVDD-PM: **We would also like to emphasize that this approximation is widely accepted and shown to be empirically successful in current related literature of diffusion models ([1,2,3]), such as classifier guidance variants when rewards are classifiers**. We will add more ablation studies on the value function quality, including SVDD-PM.
>
>
> [1] Chung, H., J. Kim, M. T. Mccann, M. L. Klasky, and J. C. Ye (2023). Diffusion posterior sampling for general noisy inverse problems. ICLR
>
> [2] Ho, J., T. Salimans, A. Gritsenko, W. Chan, M. Norouzi, and D. J. Fleet (2022). Video diffusion models. Advances in Neural Information Processing Systems 35, 8633–8646.
>
> [3] Bansal, A., H.-M. Chu, A. Schwarzschild, S. Sengupta, M. Goldblum, J. Geiping, and T. Goldstein (2023). Universal guidance for diffusion models. In Proceedings of the IEEE/CVF Conference on Computer Vision and Pattern Recognition,
>
> > W3: Incosisnte values of $\alpha$.
>
> We agree that this part should be carefully considered.  We will update our experiments to use consistent values of $\alpha$ across SVDD and baseline methods.
>
> **How diversity is retained with $\alpha=0**: While we recognize its importance, we intend to refrain from claiming our goal is to retain diversity because our primary goal is to optimize rewards while maintaining naturalness.  However, even if $\alpha=0$, it is expected that the diversity is not lost because it is regarded that we are sampling from many modes on the distribution of $p_{pre}(x)\exp(r(x)/\alpha)$ (with small $\alpha$ but not $0$ exactly) due to the randomness coming from pre-trained models and finite $M$ in practice. Indeed, we observe it as we show in molecules and images. Furthermore, as shown in Table 3, we report diversity metrics for molecular domains, e.g., Uniqueness percentage and Novelty percentage.

---

> > ### Author Response · Authors · 2024-11-29
> > **(Continue)**
> >
> > > Q1. Number of Seeds Used for Experiments
> >
> > We used 3 random seeds for all experiments. This information will be included in the revised manuscript.
> >
> > > Q2. Relation Between Consistency of Nested SMC and Number of Particles
> >
> > The number of particles M directly impacts the value function estimation quality and the sampled distribution's diversity. As M increases, the Monte Carlo estimate of the soft value function becomes more accurate, reducing bias. Meanwhile, larger M improves sampling quality but increases computational cost. We have ablation studies on the effect of M (Figure 3 and 7) to quantify this trade-off.
> >
> > > Q3. Computational Cost of Soft-Value Function Training in SVDD-MC
> >
> >  Training the soft value function in SVDD-MC involves forward passes through the diffusion model for multiple particles. While this introduces additional cost, it is significantly cheaper than fine-tuning-based methods, as it avoids modifying the diffusion model. We will add the training time.
> >
> > > Q4. Metrics for Diversity and Naturalness
> >
> > For molecular domains, as shown in Table 3, we report diversity and validity metrics, e.g., Validity percentage, Uniqueness percentage, and Novelty percentage. These results demonstrate that SVDD maintains diversity and naturalness comparable to baselines while achieving higher rewards. We will include more naturalness metrics to evaluate sample quality in more domains.
> >
> > **Regarding diversity, while we recognize higher diversity is better, representative papers about alignment in diffusion models ([1], [2], [3]), like our work, don’t have diversity metrics.** This is because alignment (reward maximization) is a primary goal, and diversity is a secondary objective. Instead, people often show generated samples to show the diversity we did. As another reason, we believe this is often a subjective metric very suspect to the definition of distance.
> >
> > [1] Clark, K., P. Vicol, K. Swersky, and D. J. Fleet (2024). Directly fine-tuning diffusion models on differentiable rewards. arXiv preprint ICLR
> >
> > [2] Fan, Y., O. Watkins, Y. Du, H. Liu, M. Ryu, C. Boutilier, P. Abbeel, M. Ghavamzadeh, K. Lee, and K. Lee (2023). DPOK: Reinforcement learning for fine-tuning text-to-image diffusion models. arXiv preprint NeurIPS
> >
> > [3] Black, K., M. Janner, Y. Du, I. Kostrikov, and S. Levine (2024). Training diffusion models with reinforcement learning

---

> > > ### Comment · Reviewer_v5um · 2024-12-02
> > >
> > > I appreciate the authors' kind response and acknowledge their various points. Given that the rebuttal confirms that SVDD should in theory sample from the target distribution presented in Section 3.2 of the paper, I still have concerns about the inconsistency in the setting of $\alpha$ across baselines leading to an evaluation biased towards SVDD. Because of this I maintain my original score, though I hope the authors revise their manuscript with updated experiments as I think the ideas in the manuscript are interesting and would be curious to see the results.

---

### Official Review · Reviewer_tbvq · 2024-11-09

**Soundness:** 3
**Presentation:** 3
**Contribution:** 2
**Rating:** 5
**Confidence:** 2

**Summary:**

This paper proposes a method for diffusion models to sample data that is both within target distribution and maximizing some downstream reward function. The problem the paper studies is of great importance, and the method shows empirical effectiveness in some downstream tasks.

**Strengths:**

- The paper is generally well-written, and the motivation is also clear. It starts with an important problem and proposes a well-motivated solution that requires no finetuning or differential proxy model.

- The paper is clear about how two critical challenges (the soft-value function is both unknown and unnormalized) are addressed by the proposed algorithm.

**Weaknesses:**

- The soft value function seems to difficult to approximate in general. Is there any anlysis or justification to quantify the quality of the approximation? How does one know a good approximation is indeed attained? Moreover, how does the approximation quality matter for the generation? More ablation study can improve the paper further.

- Is there any additional computational overhead for the proposed method? Is the approximation to the soft value function costly?

- The performance gain does not seem to be very significant compared to simple baselines, say Best-of-N. From Table 2, Best-of-N baseline is only incrementally worse than the proposed method in molecule-related experiments.

- A minor question: Does the size of the diffusion model affect the performance of SVDD? I will be interested to see how this method works for diffusion models of different size.

**Questions:**

See the weakness section.

---

> ### Author Response · Authors · 2024-11-28
>
> We appreciate your feedback.  We have addressed your concerns by explaining more about (1) the approximation quality of value functions, (2) the computational overhead of value function learning, and (3) the superior performance of our algorithm over the baseline.
>
> > *Q. The soft value function seems to difficult to approximate in general. Is there any analysis or justification to quantify the quality of the approximation? How does one know a good approximation is indeed attained? Moreover, how does the approximation quality matter for the generation?*
>
> - **Approximation error of SVDD-MC**: We evaluate the approximation of the soft value function through the learning loss, as well as its impact on the downstream reward maximization tasks. **Specifically, in Figure 6 in the Appendix, we show the training curves of value functions in SVDD-MC.** We know a good approximation is attained when the learning loss of the value function converges to a lower MSE. The consistent performance improvements across multiple domains also serve as an empirical validation of the approximation quality.
>
> - **Approximation of SVDD-PM**: We will add more ablation studies on the value function quality, including SVDD-PM. **We would also like to emphasize that this approximation is the golden standard in current related literature ([1,2,3]), such as classifier guidance variants when rewards are classifiers.**
>
> [1] Chung, H., J. Kim, M. T. Mccann, M. L. Klasky, and J. C. Ye (2023). Diffusion posterior sampling for general noisy inverse problems. ICLR
>
> [2] Ho, J., T. Salimans, A. Gritsenko, W. Chan, M. Norouzi, and D. J. Fleet (2022). Video diffusion models. Advances in Neural Information Processing Systems 35, 8633–8646.
>
> [3] Bansal, A., H.-M. Chu, A. Schwarzschild, S. Sengupta, M. Goldblum, J. Geiping, and T. Goldstein (2023). Universal guidance for diffusion models. In Proceedings of the IEEE/CVF Conference on Computer Vision and Pattern Recognition,
>
> > *Q. Is there any additional computational overhead for the proposed method? Is the approximation to the soft value function costly?*
>
> A. We acknowledge that there is additional computational overhead, primarily due to the need for multiple forwards passes through the pre-trained diffusion model to estimate the value function. We have discussed this in Sections 5.3 and Section 7. Here is a summary.
>
> - **Inference computational overhead**: As noted in Section 5.3, the computational complexity increases linearly with M (the number of samples), while memory requirements depend on whether computations are parallelized. However, compared to the Best-of-N baseline, our method is significantly more efficient under the same computational and memory budgets. **Figures 3c and 3d illustrate that SVDD achieves better results while maintaining manageable overhead.**
> - **Training computational overhead**: Our SVDD-PM variant eliminates additional training, reducing overhead when non-differentiable feedback is available. In contrast, SVDD-MC does require computational overhead; however, it learns from the reward function, which requires less computation than the fine-tuning of the pre-trained diffusion model.
>
> > *Q The performance gain does not seem to be very significant compared to simple baselines, say Best-of-N. From Table 2, Best-of-N baseline is only incrementally worse than the proposed method in molecule-related experiments.*
>
> A. Overall, the performance improvements are significant and consistent across various domains. While the performance differences may appear incremental in some domains/metrics, they represent substantial improvements in practical settings. **Table 2 shows consistent top 10% quantile improvements, which is critical for high-reward applications such as drug discovery.**
>
> > *Q. A minor question: Does the size of the diffusion model affect the performance of SVDD? I will be interested to see how this method works for diffusion models of different sizes.*
>
> A. Thank you for this suggestion. While we did not explicitly explore model size variations in this work, our method relies on inference-time computations without modifying the pre-trained diffusion model, making it inherently scalable to larger models. For future work, investigating the effect of diffusion model size on SVDD's performance is a valuable direction. We hypothesize that larger models with richer representations would further enhance the quality of the soft value function, potentially amplifying performance gains.

---

### Meta-Review · Area_Chair_EmFL · 2024-12-21

**Metareview:**

This work presents a unified framework for guidance in diffusion models, encompassing both discrete and continuous settings, with minimal additional training. It extends applicability to domains where downstream rewards may not be differentiable. The proposed method, SVDD (MC and PM), addresses discrete diffusion scenarios where continuous energy gradients cannot be directly applied to the discrete state space. Additionally, it is well-suited for cases involving non-differentiable rewards, which frequently arise in scientific domains.

However, reviewers have raised concerns regarding the novelty of the approach, noting limited differentiation from existing twisted Sequential Monte Carlo (SMC) methods. The authors also acknowledge that the term "reward maximization" may be misleading in this context. Unlike SVDD, SMC methods require resampling across the entire batch, complicating parallelization.

**Additional Comments On Reviewer Discussion:**

Despite these considerations, reviewers remain concerned about inconsistencies in baseline settings, which may bias evaluations in favor of SVDD. Consequently, the initial rating remains unchanged after rebuttal.

---

### Decision · Program_Chairs · 2025-01-22

Reject